# Enhancing photosynthetic CO₂ fixation by assembling metal-organic frameworks on *Chlorella pyrenoidosa*

Dingyi Li [1,4], Hong Dong [2,4], Xupeng Cao [2,3], Wangyin Wang [2] ✉ & Can Li [1,2,3] ✉

The CO₂ concentration at ribulose-1,5-bisphosphate carboxylase/oxygenase (Rubisco) is crucial to improve photosynthetic efficiency for biomass yield. However, how to concentrate and transport atmospheric CO₂ towards the Rubisco carboxylation is a big challenge. Herein, we report the self-assembly of metal-organic frameworks (MOFs) on the surface of the green alga *Chlorella pyrenoidosa* that can greatly enhance the photosynthetic carbon fixation. The chemical CO₂ concentrating approach improves the apparent photo conversion efficiency to about 1.9 folds, which is up to 9.8% in ambient air from an intrinsic 5.1%. We find that the efficient carbon fixation lies in the conversion of the captured CO₂ to the transportable HCO₃⁻ species at bio-organic interface. This work demonstrates a chemical approach of concentrating atmospheric CO₂ for enhancing biomass yield of photosynthesis.

Photosynthesis in biosphere involves a fundamental process of initial CO₂ capture and fixation. The overall photosynthetic efficiency depends on both the efficiencies of solar-to-chemical energy conversion in light reactions, and the carbon fixation in dark reactions[1–4]. The insufficient supplementation and low CO₂ affinity with ribulose-1,5-bisphosphate carboxylase/oxygenase (Rubisco) limit the process of carbon fixation in chloroplast[5,6]. Hence, besides improving the activity of Rubisco[7,8], it's also important to understand the carbon concentrating mechanism (CCM) in photosynthetic organisms to develop more efficient CO₂ fixation methods[9–11]. Cyanobacteria and eukaryotic algae in the biosphere contribute to the majority of global carbon assimilation and oxygen evolution[12–15]. Different from CCM in terrestrial C₄ plants which rely on atmospheric CO₂ uptake and special Kranz anatomy derived from cell differentiation[16], aquatic microalgae operate CCM in a unicellular manner. Between two major forms of dissolved inorganic carbons (DIC) in water, non-polar CO₂ molecules tend to diffuse across biomembrane driven by concentration gradient, while HCO₃⁻ relies on ATP-dependent transportation of membrane-bound transporters and is enriched in microalgal cell[17]. It was recently reported that mitochondrial-produced ATP energizes HCO₃⁻ transporters on cytoplasm and chloroplast envelope in the green alga *Chlamydomonas reinhardtii*[18]. Carbonic anhydrase (CA) in periplasm and chloroplast plays a crucial role in the interconversion of two major forms of DIC, CO₂, and HCO₃⁻, which strengthens the directed HCO₃⁻ active transportation to the thylakoid, then converts HCO₃⁻ to CO₂ in the proximity of Rubisco at a low luminal pH generated by alternative photosynthesis pathways, and recaptures inevitably leaked CO₂ to HCO₃⁻ in chloroplast stroma[18,19]. When ambient CO₂ supplementation is not enough, the CCM in algae is commonly activated to compensate the constraint of insufficient carbon supply from ambient atmosphere[20]. Although CCM is prone to be manipulated for increasing carbon fixation, the effectiveness is difficult to compensate the constraint of insufficient carbon supply from ambient atmosphere. Therefore, intensifying capture of CO₂ dissolved in the solution to increase the concentration of CO₂ around the algal cell is an alternative approach to the transport of CO₂ into the cell for improving photosynthetic efficiency.

Metal-organic frameworks (MOFs) composed of metal nodes and organic ligands are promising materials for CO₂ capture

[1]School of Materials Science and Engineering, National Institute for Advanced Materials, Nankai University, Tianjin, China. [2]State Key Laboratory of Catalysis, Dalian Institute of Chemical Physics, Chinese Academy of Sciences, Dalian National Laboratory for Clean Energy, Dalian, China. [3]University of Chinese Academy of Sciences, Beijing, China. [4]These authors contributed equally: Dingyi Li, Hong Dong. ✉e-mail: wywang@dicp.ac.cn; canli@dicp.ac.cn

and storage from ambient air[21–24]. Appropriate humidity is reported to be beneficial to $CO_2$ adsorption in microporous and mesoporous MOF, as pre-adsorbed $H_2O$ molecules in mesopores can form microporous pockets to enhance the $CO_2$ confinement at low pressures[25–28]. Moreover, amine-functionalized MOF exhibits enhanced $CO_2$ uptake capacity in humid conditions benefits from the high affinity of amino groups to $CO_2$[26]. However, it is not reported that the assembly of such material on biological system would be favorable to $CO_2$ capture and transportation to microalgal cells in aqueous solution.

Here, we demonstrate an artificial $CO_2$ concentrating approach by self-assembly of MOFs on microalgae for enhancing algal photosynthetic efficiency. The chemical concentrating $CO_2$ approach improves the apparent photo conversion efficiency to about 1.9 folds, which is up to 9.8% of the MOF/*C. pyrenoidosa* in ambient air. We find that the efficient conversion of the enriched $CO_2$ to bicarbonate by the excreted carbonic anhydrase of *C. pyrenoidosa* intensifies the $CO_2$ supply to microalgae and stimulates the over-expression of Rubisco, which results in the enhancement of biomass production.

## Results
### Self-assembly of MOF on *C. pyrenoidosa*

The as-prepared MOF material has an average size of 300 nm (Supplementary Fig. 1), good crystallinity (Supplementary Fig. 2), and high BET surface area of 1143 $m^2 g^{-1}$ (Supplementary Fig. 3, Supplementary Method 1). It is self-assembled with *C. pyrenoidosa*, a kind of well commercialized green alga. $CO_2$ adsorption property of the MOF was investigated in both gas and liquid phases. Figure 1a shows the relationship between $CO_2$ adsorption/desorption and relative pressure of $CO_2$. The capacity of MOF for $CO_2$ adsorption is estimated to be as high

as 20 $cm^3 g^{-1}$. Figure 1b shows that the $CO_2$ breakthrough curves of the MOF in ambient air (400 ppm $CO_2$) under different relative humidity (RH). The adsorption capacity of $CO_2$ is estimated to be 1.3 $mg\,g^{-1}$ under 3% RH and 1.8 $mg\,g^{-1}$ under 45% RH, indicating that water molecule has a stimulating effect on $CO_2$ adsorption. Figure 1c displays the $CO_2$ adsorption of the MOFs in aqueous solution. It can be seen that the adsorption amount of DIC ($CO_2$ and $HCO_3^-$) increases linearly with the MOF loading and complete adsorption is achieved when the amount of MOF is reached to 100 $mg\,L^{-1}$. It is noted that $CO_2$ can be also desorbed in aqueous solution (Supplementary Fig. 4). These results indicate that the MOF has capability of $CO_2$ enrichment in microalgal culture/growth medium.

Matching of surface charge between MOF and algae is important for self-assembly. Figure 1d shows the Zeta potential changes at different pH values. It can be seen that the MOF possesses positive surface charge in a broad pH range from 6 to 9 in the culture conditions. It is known that the surface of *C. pyrenoidosa* cell is negatively charged, because the cell wall is formed by a microfibrillar layer composed of polysaccharides and proteoglycans[29,30]. Consequently, the opposite electrostatic potentials between MOF and *C. pyrenoidosa* tend to drive them self-assembly by van der Waals interaction as supported by flow cytometric results (Supplementary Fig. 5). We also noticed that the MOF on the surface of *C. pyrenoidosa* protects photosystem II from photodamage by strong light or UV stress rather than blocking the light harvesting properties of microalgal chloroplast (Supplementary Fig. 6–8).

The self-assembled structure of MOF on *C. pyrenoidosa* were examined by microscopic analysis. Figure 2a shows that the MOF material is in octahedral shape, while Fig. 2b shows that the *C. pyrenoidosa* cell has a spherical surface with an average diameter of 3 μm

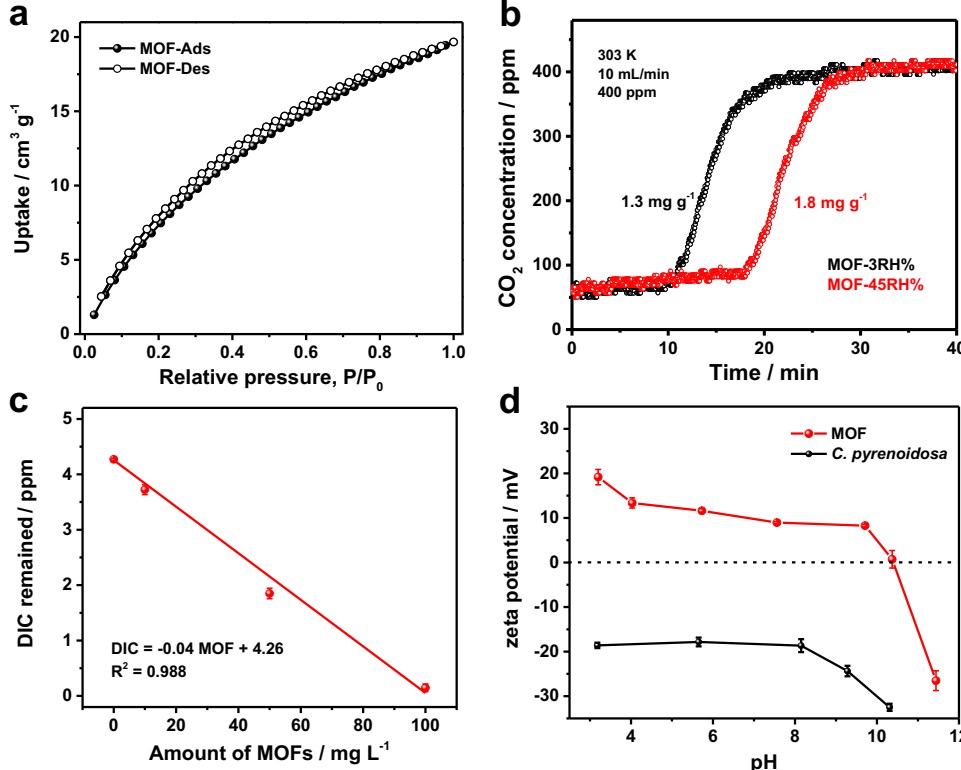

**Fig. 1 | $CO_2$-capture performances of MOF. a** The $CO_2$ adsorption/desorption isotherms of MOF (NH$_2$-MIL-101-Fe) at 298 K. **b** Competitive $CO_2$ breakthrough curves measured at two relative humidity (RH) values (Temperature, 303 K; Flux, 10 mL/min; $CO_2$ concentration, 400 ppm; MOF amount, 50 mg). **c** Plots of the remaining DIC concentration in air-saturated BG-11 medium versus the additional

amount of MOF. (Temperature, 303 K; pH, 7.0). **d** Plot of the zeta potentials of MOF and *C. pyrenoidosa* versus the pH values of cultivation medium. Error bars in **c**, **d** represent the standard deviation of the results from three biologically independent experiments. Source data are provided as a Source Data file.

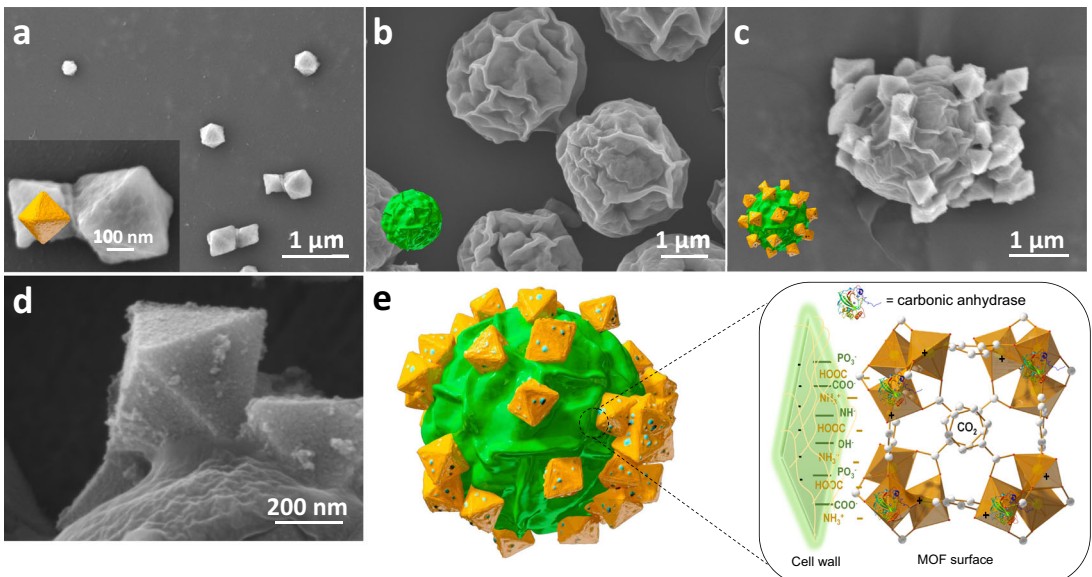

**Fig. 2 | Self-assembly of MOF on the surface of *C. pyrenoidosa*.** SEM images of **a** MOF in fresh BG-11 medium, **b** *C. pyrenoidosa*, **c**, **d** the hybrids of MOF and *C. pyrenoidosa* of **c** low and **d** high magnification, **e** Proposed self-assembly models of MOF/Algae hybrid.

(Supplementary Fig. 9). Figure 2c is the SEM image of the MOF/*C. pyrenoidosa* hybrid sample, which clearly shows that the MOF particles are adsorbed on the surface of the *C. pyrenoidosa* cell. Interestingly, Fig. 2d shows the MOF particles adsorbed on the cell become relatively tight and uniformly distributed after culturing for two days (Supplementary Fig. 10). In addition, several small particles are dispersed on the surface of the MOFs in the culture supernatant of *C. pyrenoidosa* after cultivation (Supplementary Fig. 11). It indicates that the free enzymes are diffused out of the cell and adsorbed on the MOF. Figure 2e illustrates the entire self-assembled hybrid of MOF with microalga. The interfacial interactions between MOF and *C. pyrenoidosa* cell, including electrostatic attractions, Van der Waals forces and hydrogen bonding, are accountable for the self-assembly of MOF on *C. pyrenoidosa* cell.

## Biomass growth in the MOF/*C. pyrenoidosa*

Cell growth was investigated and the dry cell weight was used to evaluate the performance of carbon fixation (Supplementary Fig. 12). Figure 3a shows the relations between microalgal biomass growth and the amount of MOF in the microalgal cultivation medium under ambient air. It can be seen that the biomass production rate of bare *C. pyrenoidosa* is at a rate of 0.13 g L$^{-1}$ day$^{-1}$. Interestingly, the biomass production rate is significantly increased when MOF was adsorbed on the *C. pyrenoidosa* algae. When 50 ppm MOF is used for the self-assembly with *C. pyrenoidosa*, the biomass growth rate reaches the maximum value with a rate of 0.25 g L$^{-1}$ day$^{-1}$, in which is about 1.9-fold enhancement of biomass growth. Further increasing the loading amount of MOF to 100 ppm doesn't accelerate the biomass growth, indicating that the adsorbed MOF particles could be saturated for *C. pyrenoidosa* cells. Figure 3b shows the influence of pH value on the growth of MOF/*C. pyrenoidosa* hybrid system. Neutral and weak alkaline environments (pH 7, 8) are preferable for microalgal growth than weak acidic environment (pH 6). Under pH 7 condition, the MOF/*C. pyrenoidosa* hybrid considerably increases the biomass growth. Similar biomass growth promotion phenomenon was also observed on another green alga *Chlamydomonas reinhardtii* (Supplementary Fig. 13). As the pH value commonly dominates the form of $CO_2$ in aqueous solution, and the biophysical reaction is closely related to the proton gradient, the pH drift level of the above different systems was examined. Figure 3c displays the pH drift level of

the MOF/algae system during the culture process. It can be seen that the pH value drifts to alkaline direction for MOF/algae system, which is more pronounced than that of the control at pH range from 6 to 8. The pH drift to alkaline indicates that $OH^-$ is released to the outside of the cell after converting $HCO_3^-$ into $CO_2$ for fixation. The distinct alkaline microdomain at interface between MOF and algae is prone to promote the conversion of the captured $CO_2$ into $HCO_3^-$. The above observation can serve as the sign of the $HCO_3^-$ utilization capacity of microalgae, which positively correlates with the microalgal biomass growth[31]. Figure 3d exhibits the microalgal quantum yield of PS II during cultivation. The change on photosynthetic pigments after adding MOF was also measured (Supplementary Fig. 14). The result shows that the performance of the light reactions in the assembled MOF/algae system is the same as that of the bare algae even at different pH points. Hence, the biomass growth in hybrid system is contributed to the dark reaction of carbon fixation. According to the optimized results of biomass production, the apparent photo conversion efficiency of the MOF/algae system is calculated to be 9.8% (Supplementary Fig. 15).

The chemo-stability of MOF during long-term cultivation of *C. pyrenoidosa* was tested to exclude the interference of organic carbon for algae biomass growth. About 13% organic ligand of MOF was released into the culture medium after two-day cultivation, but the free organic ligand and Fe ion hardly accelerated the growth of *C. pyrenoidosa* (Supplementary Fig. 16). In order to see if it was the porous nature of MOF functions as $CO_2$ adsorber and concentrator contributing to algal photosynthesis, some other typical porous materials, MIL-101-Fe, MIL-101-Cr and a commercial zeolite 13X, were also tested in the same experimental conditions. All these materials show a similar promotion behavior for algal photosynthetic carbon fixation (Supplementary Fig. 17). These results reveal that the MOF or zeolite assembled on the surface of the algae cell enables $CO_2$ enrichment and transportation for photosynthetic carbon fixation in algae.

## $CO_2$ transportation mechanism from MOF to *C. pyrenoidosa*

To understand how the $CO_2$ molecule is adsorbed on the MOF and then transported into the cell, we first investigate the carbonic anhydrase (CA) activity of the *C. pyrenoidosa* using two inhibitors, acetazolamide (AZA) and ethoxazolamide (EZA), which can block the $CO_2$

transport chain. The carbonic anhydrases in the periplasm (external CA, eCA) and the chloroplast (internal CA, iCA) are both responsible for the regulation of the balance between $CO_2$ and $HCO_3^-$ in total DIC[32–34]. It is known that the membrane-impermeable AZA inhibitor targets on the inhibition of periplasmic eCA[35]. While the membrane-permeable EZA inhibitor targets on both eCA and iCA as it is a kind of comprehensive inhibitor to CCM[36].

Figure 4a shows the biomass growth of *C. pyrenoidosa* and MOF/ *C. pyrenoidosa* cells grown at different pH under air in the presence or absence of 1 mM eCA inhibitor AZA. The addition of AZA shows slight effect on the biomass growth of both *C. pyrenoidosa* and MOF/ *C. pyrenoidosa* cells at pH 6. But the addition of AZA causes considerable decrease of biomass growth rate of MOF/*C. pyrenoidosa* cells at pH 7 and 8. After treated with the EZA inhibitor, the biomass growth of *C. pyrenoidosa* was almost stopped when EZA was used (Supplementary Fig. 18). This indicates that AZA inhibits the extracellular hydration of $CO_2$ to $HCO_3^-$ catalyzed by eCA and EZA inhibits the intracellular dehydration of $HCO_3^-$ to $CO_2$ catalyzed by iCA for CBB cycle which is vital to microalgal cell growth. The free eCA can diffuse into the medium through the cell wall for hydration of the dissolved $CO_2$ in culture medium[37,38]. The inhibition of eCA gives rise to the sharp decline of biomass growth of the MOF/algae system. On the other hand, the activity of eCA left in medium are assayed at different pH to understand the process of CCM (Fig. 4b). The activities are nearly the same at pH 7 and 8, but the activity of CA at pH 6 decreased by 42%. Due to the adsorption of CA on the surface of MOFs, the apparent activities of CA left in medium are reduced correspondingly. It also can be ascribed to the improved DIC uptake with MOF lowering the expression level of eCA serving as a low $CO_2$ inducible enzyme. With pH increasing from 6 to 8, CA activity in the media of low-$CO_2$ grown *C. pyrenoidosa* suspensions also increases due to the favorable environment for the hydration of $CO_2$ to $HCO_3^-$. (Supplementary Fig. 19).

The adsorbed protein amount and CA activity on MOF particles were measured (Supplementary Fig. 20). After the activities normalized by protein amount, the specific activity of CA on MOFs is similar with that in free condition, indicating that eCA adsorbed on MOF enables efficient conversion of MOF-captured $CO_2$ into $HCO_3^-$. The result is consistent with the image in Fig. 2e where most of the eCAs released by cell were adsorbed on the MOF, resulting in the promotion of the DIC supplementation to *C. pyrenoidosa* and hence acceleration of biomass growth (Supplementary Fig. 21).

In order to support the mechanism that MOF acts as a $CO_2$ concentrator to accelerate the process of photosynthetic carbon fixation, we measured the apparent affinity of net $O_2$ evolution for $CO_2$ concentration of *C. pyrenoidosa* and MOF/*C. pyrenoidosa* cells cultivated at different pH (Supplementary Fig. 22). Figure 4c shows half saturation constant ($K_{1/2}$) of the system to indicate the apparent affinity for $CO_2$. We observe that the $K_{1/2}$ for $CO_2$ of *C. pyrenoidosa* cells is 65 μM at pH 6, decreases to 31 μM at pH 7 and increases to 43 μM at pH 8. And under all the conditions from pH 6 to pH 8, the function of MOF enables the $K_{1/2}$ for $CO_2$ decreased comparatively. Especially, MOF particles enables the apparent affinity for $CO_2$ enhanced by 82% at pH 7, indicates the synergetic effect between MOF capturing $CO_2$ and the intrinsic CCM in *C. pyrenoidosa* cells. The effect of MOF obviously promotes the biomass growth increased to 1.9-fold of *C. pyrenoidosa* cells (harboring a CCM) grown at pH 7 under air (LC, 0.04%), but in the case of cells grown for high $CO_2$ (2%) without harboring a CCM, no role of MOF is played in enhancing the biomass growth (Supplementary Fig. 23). These results verifies that MOF as a $CO_2$ concentrator is favorable for accelerating $CO_2$ hydration into $HCO_3^-$ by eCA. Figure 4d shows the reaction kinetics of the hydration of $CO_2$ into $HCO_3^-$ by eCA. The eCAs adsorbed on the MOF plays a crucial role in the promotion of the DIC supplementation to *C. pyrenoidosa* and hence acceleration of biomass growth (Supplementary Fig. 24).

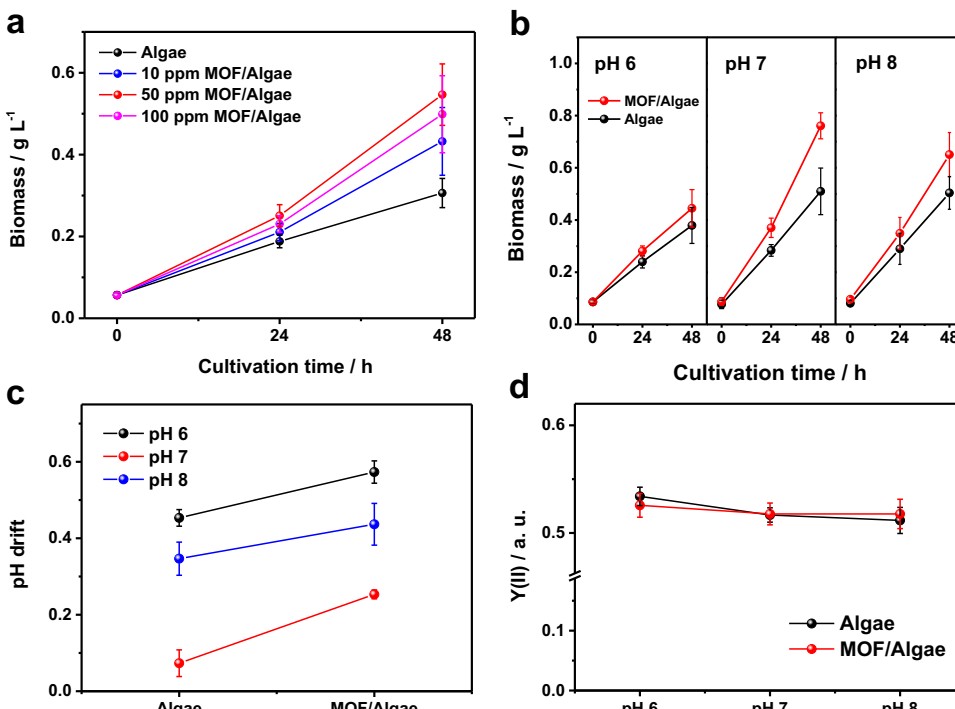

**Fig. 3 | Biomass growth in the MOF/*C. pyrenoidosa*. a** The growth curves of *C. pyrenoidosa* with the addition of different amounts of MOF at initial pH 7 and be aerated with ambient air. **b** The growth curves of *C. pyrenoidosa* and MOF/Algae during two-day cultivation in a photobioreactor containing BG-11 medium which was buffered by 20 mM HEPES with different initial pH of 6, 7 and 8. **c, d** pH-drifts of media **c** and effective PS II quantum yield, Y(II) of *C. pyrenoidosa* **d** corresponding to **b**, the white dash line in **c** marked the initial pH of media. Cultivation conditions: temperature, 26 °C; light intensity, 50 μE m$^{-2}$ s$^{-1}$; 20 mL min$^{-1}$ ambient air flow (LC); 24 h continuous illumination. Error bars in **a**–**d** represent the standard deviation of the results from three biologically independent experiments. Source data are provided as a Source Data file.

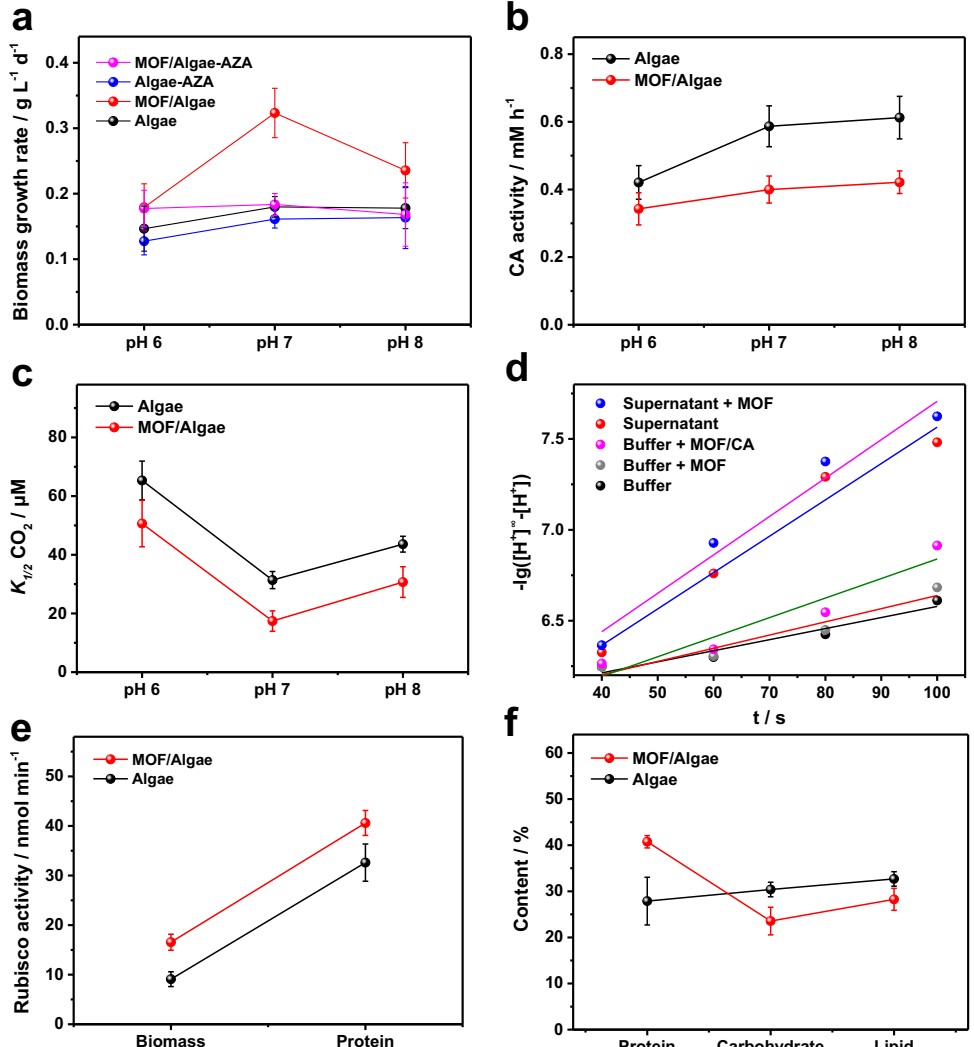

**Fig. 4 | The coupling of MOF and the intrinsic CCM of *C. pyrenoidosa*. a** The average biomass growth rate of *C. pyrenoidosa* within two-day cultivation with the addition of 50 ppm MOF, 1 mM AZA (acetazolamide, inhibit external carbonic anhydrase, eCA) and both of them at different pH. **b** CA activity measurements of the supernatants of microalgal suspensions which cultivated with or without MOF at different pH under air. **c** The Michaelis constant $K_{1/2}$ values as calculated from the Michaelis-Menten fit of the plot of the net $O_2$ evolution rate versus the concentration of $CO_2$ for *C. pyrenoidosa* (Algae) and MOF/*C. pyrenoidosa* (MOF/Algae) cells grown at different pH under air (LC, 0.04%), error bars in (**c**) represent the standard deviation of $K_{1/2}$ values as fitted from three independent experimental results. Cell density of $1.0 \times 10^7$ cell mL$^{-1}$ and the light source with a cutoff wavelength filter (under 50 µE m$^{-2}$ s$^{-1}$ irradiation, λ > 600 nm). **d** The linear fitting curves of the kinetic plots of the reaction that the hydration of $CO_2$ into $HCO_3^-$ in different

environments (pH 7.0 20 mM HEPES buffer, the supernatant of microalgal suspension after two-day cultivation, and the addition of bare MOF or MOF pre-treated in supernatant), pH changes indicate the accumulation of products since proton generated simultaneously with $HCO_3^-$ in this process. **e** The apparent activities of the Rubisco based on the dry cell weight of biomass and the total protein, respectively. **f** The protein, carbohydrate and lipid contents of *C. pyrenoidosa* cultured two days in the absence (Algae) and presence (MOF/Algae) of MOF. Cultivation conditions: temperature, 26 °C; light intensity, 50 µE m$^{-2}$ s$^{-1}$; 20 mL min$^{-1}$ ambient air flow (LC); 24 h continuous illumination; 20 mM HEPES buffer (initial pH 7.0). Error bars in **a**–**f** represent the standard deviation of the results from three biologically independent experiments. Source data are provided as a Source Data file.

To understand the carboxylation of the Rubisco enzyme in the process of carbon fixation, activities of the Rubisco enzyme based on the amount of total biomass and protein were tested. Figure 4e shows that the apparent activity of the Rubisco increased from 9.1 to 16.5 nmol min$^{-1}$ per mg biomass, and from 32.6 to 40.6 nmol min$^{-1}$ per mg protein. The expression level of Rubisco qualitatively increased (Supplementary Fig. 25). Figure 4f shows the contents of proteins, carbohydrates and lipids in dry cell weight. Among them, the protein content increases from 28% to 41% after the addition of MOF, while the carbohydrate decreases from 30% to 24% and the lipid decreases from 32% to 27%, which were calculated by the analysis methods (Supplementary Fig. 26). These results reveal that such an increase in Rubisco content may result from an increased internal

Ci concentration in MOF-treated algae, and may contribute to reaching high $CO_2$ fixation rates and high biomass productivity. However, no increase in the maximal $O_2$ evolution rate was observed under non-limiting $CO_2$ (Supplementary Fig. S22, S27), indicating that the increased Rubisco content in MOF-treated algae mainly contributes to dark reactions.

## Discussion
In this study, we find that the affinity for $CO_2$ of MOF/*C. pyrenoidosa* cell is higher than that of bare *C. pyrenoidosa* cell (Supplementary Fig. 27). But the presence of MOF particles doesn't remarkably have influence on the affinity of cells grown under high $CO_2$ (Supplementary Fig. 27). Because *C. pyrenoidosa* cells grown in high $CO_2$ is not

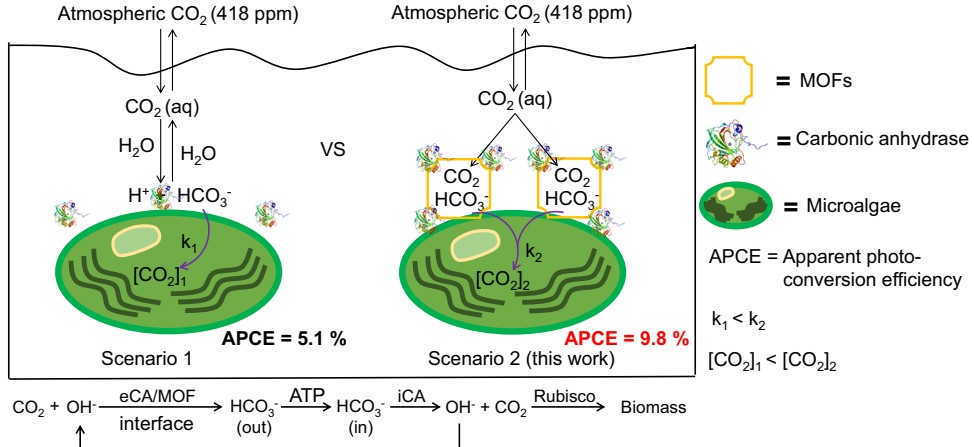

**Fig. 5 | The proposed CO₂-enrichment and transport mechanisms of the MOF/*C. pyrenoidosa* assembly.** Scenario 1: natural carbon fixation of microalgae growth is limited by the $CO_2$ concentration in the cell ($[CO_2]_1$) and enzymatic kinetic rate $k_1$. Scenario 2: in this work, MOFs as a $CO_2$ concentrating entrepot are applied to accelerate the transport of $CO_2$. The captured $CO_2$ in MOF is hydrated to $HCO_3^-$ by the excreted eCA adsorbed on MOF. The higher concentration of $HCO_3^-$ outside is transported into the cell by the transporter, and then in pyrenoid, $HCO_3^-$ is converted to $CO_2$ by iCA to feed the carboxylation process of Rubisco. Compared to the intrinsic carbon concentrating mechanisms that hydrate dissolved $CO_2$ to $HCO_3^-$ for inorganic carbon supplementation, the artificial $CO_2$-enrichment and transport pathways not only increase the $CO_2$ concentration ($[CO_2]_2$) in the cell but also accelerate the rate of Rubisco ($k_2$) for $CO_2$ fixation, which increases the apparent photo conversion efficiency to 1.9 folds, up to 9.8% in ambient air.

harboring the intrinsic CCM[20]. The complete CCM is vital to transfer $CO_2$ toward Rubisco enzyme for accelerating the rate of $CO_2$ fixation. Moreover, we find that the expression level of Rubisco protein is upregulated in MOF/*C. pyrenoidosa* cells (Fig. 4e). It was reported that the Rubisco content in *C. pyrenoidosa* was affected by the $CO_2$ concentration during cultivation and nearly full activity of Rubisco must be needed during photosynthesis in *C. pyrenoidosa* and other green algae[39]. It was observed that Rubisco protein is more likely to stay in the chloroplast stroma than to be close-packed in the pyrenoid when $CO_2$ concentration is elevated. And the additional Rubisco was distributed in the chloroplast stroma rather than to be organized in the pyrenoid[40]. The kinetic properties of Rubisco vary among different photosynthetic organisms, but remain steady on a certain species[13]. It implies the biomass growth rate strong dependence on the Rubisco content in *C. pyrenoidosa*. Thus, the synergetic effect of the functional MOF and the intrinsic CCM in *C. pyrenoidosa* cells enables a high affinity for $CO_2$ and the Rubisco content in *C. pyrenoidosa* for accelerating $CO_2$ fixation.

Figure 5 illustrates the $CO_2$ enrichment and transportation mechanisms of MOF/*C. pyrenoidosa* assembly under atmospheric $CO_2$ conditions. In artificial part, MOF captures and concentrates $CO_2$ from the culture medium balanced with the air. The eCAs adsorbed on MOF catalyzes the captured $CO_2$ hydration to $HCO_3^-$. The increasing amount of $HCO_3^-$ at bio-inorganic interfacial microdomain intensifies the transportation of $HCO_3^-$ into the cell. the artificial $CO_2$-enrichment and transport pathways not only increase the $CO_2$ concentration in the cell but also accelerate the kinetic rate of Rubisco for $CO_2$ fixation. In natural part, $HCO_3^-$ in periplasm is transported via the membrane-bound $HCO_3^-$ transporters to chloroplast pyrenoid. Finally, $HCO_3^-$ ions are converted to $CO_2$ by iCA in pyrenoid to feed the carboxylation process of Rubisco (Supplementary Fig. 28). The reversible switch between $HCO_3^-$ and $CO_2$ catalyzed by eCA and iCA enzymes plays a crucial role in driving $CO_2$ delivery to the Rubisco sites for $CO_2$ fixation reaction. After $HCO_3^-$ conversion into $CO_2$ by iCA, the alkaline microdomain formed at the interface between MOF and alga promotes the captured $CO_2$ conversion into $HCO_3^-$ by eCA enzyme. Therefore, bio-organic interfacial microenvironmental proton gradient and related ATP production are attributed as the driving force for extracellular $CO_2$ capture and conversion to $HCO_3^-$ by the MOF/*C. pyrenoidosa* for carbon fixation to enhance biomass production.

In brief, a MOF/algae self-assembled hybrid system is constructed through the adsorption of MOF particles on the surface of the *Chlorella pyrenoidosa* for efficient photosynthetic $CO_2$ fixation. $CO_2$ enrichment on MOF/*C. pyrenoidosa* assembled system increases the biomass growth of *C. pyrenoidosa* by 1.9-folds compare with that of bare *C. pyrenoidosa*, and elevates the expression level of Rubisco by 82% meantime. The apparent photo conversion efficiency reaches up to 9.8% with the aid of the artificial CCM. The mechanism study shows that $CO_2$ is firstly captured by MOFs and then converted into $HCO_3^-$ by the extracellular eCA on the surface of MOF for catalytic $CO_2$ hydration at bio-inorganic interface between the artificial and natural counterparts. $HCO_3^-$ is confirmed to be the main form of inorganic carbon source for transportation in *C. pyrenoidosa*, elucidating the well coupling between the MOF-based artificial CCM and the intrinsic CCM of *C. pyrenoidosa*. The artificial concentration of $CO_2$ by the assembly of algae with MOF reported in this work provides an avenue for improving ambient $CO_2$ supplementation and facilitating algal photosynthesis for more efficient $CO_2$ fixation and conversion.

## Methods
### Algal culture
*Chlorella pyrenoidosa* (FACHB-9) was cultured with 40 mL BG-11 medium in an air-lift column photobioreactor (20 cm high and 2 cm in diameter) under 50 μE m⁻² s⁻¹ continuous one-side illumination from a white fluorescent lamp (temperature, $26 \pm 2$ °C). Ambient air (LC) or 2% $CO_2$ enriched air (HC) was set as 20 mL min⁻¹. About 0.1 g L⁻¹ cells were inoculated and harvested after two days cultivation by centrifugation (2220 g, 5 min). For MOFs and other $CO_2$-captured materials assisted algal cultivation, as well as the addition of CA inhibitors, a certain amount of their suspension/solution were injected to BG-11 medium before algal inoculation. The cell density was determined by the absorbance at 750 nm ($OD_{750}$) and calibrated by a calibration curve of dry cell weight (DCW) versus $OD_{750}$. 20 mM 4-(2-hydroxyethyl)1piperazineethanesulfonic acid (HEPES) buffer solution was supplemented to adjust the initial pH of BG-11 medium if necessary. *Chlamydomonas reinhardtii* (cc-137) was cultured in the same conditions except the medium was replaced by TAP medium. Three biological replicates of each growth condition were processed.

## Synthesis of NH$_2$-MIL-101-Fe

NH$_2$-MIL-101-Fe was prepared by reaction of 2-aminoterephthalic acid (NH$_2$-H$_2$BDC) with FeCl$_3$·6H$_2$O in DMF at 110 °C for 36 h. After being cooled to room temperature, the resultant precipitates were separated by centrifugation and washed thoroughly with DMF and ethanol. Then, the as-prepared NH$_2$-MIL-101-Fe was soaked in EtOH with reflux for 24 h in order to exchange of the guest solvent molecules and excess ligand molecules. Finally, the obtained product was dried by vacuum drying.

## Flow cytometry

MOF, *C. pyrenoidosa* and MOF/*C. pyrenoidosa* were redispersed in BG-11 medium to be analyzed by a Sony SH800 flow cytometer. The cell density of *C. pyrenoidosa* was $0.2 \times 10^7$ cell mL$^{-1}$, and the concentration of MOFs is 10 ppm. The sheath fluid is 10 mM phosphate buffer solution (pH 7.2). FSC (forward scatter) mode was used for MOF and *C. pyrenoidosa* to analyze the particle size distribution of those sample suspensions according to their FSC-area signals (analyzed in Cell Sorter Software version 2.1).

## CO$_2$ adsorption/desorption performances

CO$_2$ adsorption isotherms of NH$_2$-MIL−101-Fe (298 K) were plotted by ASAP2020 physical adsorption analyzer (Micromeritics, USA) to determine the surface area and pore volume of MOF. 100 mg MOF powder in tube was degassed 2 h under high vacuum and 100 °C, and then measured its CO$_2$ adsorption isotherms at 298 K. Breakthrough experiments were performed in a column packed with 50 mg fully activated MOF powder. 10 mL min$^{-1}$ airflow with different humidity flew through the column, and the CO$_2$ concentration at the outlet of the column was monitored online with an IR detector. Temperature was controlled at 303 K. For the measurement of CO$_2$ adsorption capacity in water, 50 mg MOF was dispersed in 20 mL fresh BG-11 medium (pH 7.0 20 mM HEPES buffer), 20 mL min$^{-1}$ airflow was purged for 2 h, then separated MOF powder by centrifugation and subsequent dried in 60 °C for 12 h, and immediately thermal desorbed as breakthrough experiments at 363 K.

## Electron microscopy measurements

NH$_2$-MIL-101-Fe, *C. pyrenoidosa* and MOF/*C. pyrenoidosa* were centrifuged (2220 g, 5 min) and rinsed three times with deionized water, and subsequently dehydrated in anhydrous ethanol for 30 min. The cell suspensions in ethanol were dropped onto a copper grid (3 mm in diameter) and dried in the air. Scanning electron microscopic characterizations were conducted on a JSM-7900F FESEM (JEOL, Japan) after samples sputtered with 2 nm platinum foil.

For the observation of the cross-section of bare *C. pyrenoidosa*, the prefixed cells should be further fixed by 1% osmic acid, then dehydrated in a graded series of ethanol solutions, then embedded in epoxy resin. Finally, the samples were ultrathin sectioned at ~50 nm thickness and placed on a copper grid for transmission electron spectroscopic observation (HT7700, HITACHI, Japan).

## CA activity measurements

The enzymatic activities of extracellular carbonic anhydrase (CA) that existed in the supernatant of microalgal suspensions were determined with a colorimetric CA activity assay kit (Solarbio, China). Briefly speaking, 0.7 mL of Tris buffer (50 mM, pH 7.5), 0.2 mL of p-nitrophenyl acetate (p-NPA) aqueous solution, and 0.1 mL of medium centrifuged from different microalgal suspensions (or the redispersed suspension of MOF pretreated in medium) were mixed in the quartz cuvette. Since the hydrolysis of p-NPA to p-nitrophenol (p-NP) can be catalyzed by CA, the increase rate in the concentration of p- (p-NP) indicates the enzymatic activity of CA. The absorbances at 405 nm of the mixture were recorded before and after 5 minutes' incubation in water bath (37 °C) to calculate the CA activity. The self-

dissociation of p-NPA was measured in the same conditions for correction. The concentration of p-NP was calibrated by p-NP standard solutions.

## Reaction kinetics of the hydration of CO$_2$ into HCO$_3^-$

The reaction kinetics of the hydration of CO$_2$ into HCO$_3^-$ in the supernatant of microalgal suspension or the MOF suspension were monitored according to Rawat[37]. Briefly speaking, 2 mL of the samples (OD$_{750}$ = 2.0 for microalgal suspensions, contain 20 mM HEPES, pH 8.0) were kept at 4 °C. The control group was set as 2 mL 20 mM HEPES buffer (pH 8.0). Adding 1 mL of ice-cold CO$_2$-saturated water to initiate the reaction. Then record the time intervals (T) required for the pH drop from 7.7 to 6.3. To record the kinetics of CO$_2$ hydration, pH was recorded per 20 s during the process.

## Chlorophyll fluorescence measurements

For PSII quantum yield measurements, *C. pyrenoidosa* and *C. pyrenoidosa*/MOF were suspended in a quartz sample cuvette and recorded with Water-PAM fluorometer (Heinz Walz GmbH, Germany) with an actinic light of 226 μE m$^{-2}$ s$^{-1}$ after 10 min dark adaptation. For PSII electron transfer rate (ETR II) measurements, the actinic light intensity varies from 0 to 2405 μE m$^{-2}$ s$^{-1}$.

## Rubisco identification and activity measurements

*C. pyrenoidosa* and *C. pyrenoidosa*/MOF were harvested by centrifugation (2220 g, 5 min) for three times, then resuspended in 1 mL Rubisco extraction buffer, followed by disruption using sonication. The supernatant was collected by centrifugation (12000 g, 10 min, 4 °C). Finally, Rubisco activity of were determined with a colorimetric Rubisco activity assay kit (Solarbio, China). Briefly speaking, ribulose-1,5-biphosphate (RuBP) interacted with CO$_2$ to produce 3-phosphoglycerate (3-PGA) with the catalysis of Rubisco. 3-PGA could be reduced to 3-phosphoglyceraldehyde which was stoichiometrically fueled by NADH. By monitoring the absorbance at 340 nm in 5 min interval, the Rubisco activity was determined. For the measurement of Rubisco expression level, the protein samples above were separated on a 10% SDS/PAGE, then be stained by coomassie brilliant blue. Rubisco was identified by Western blot. The protein was separated by SDS-PAGE and then transferred onto PVDF membranes (Millipore Co., USA) in Tris-glycine buffer at 100 mA for 2 h. The membranes were blocked with 5% (w/v) nonfat milk in Tris-buffered saline containing 0.1% (v/v) Tween-20 (TBST) at 37 °C for 2 h. The membrane was then incubated with the appropriate primary antibodies (Catalog No. AG5359, Beyotime, China) at a dilution ratio of 1:10,000 at 4 °C overnight. After three washes with TBST, the membrane was incubated with horseradish-conjugated goat-anti-rabbit secondary antibody (Catalog No. A0208, Beyotime, China) at a dilution ratio of 1:3000 for 1 h at room temperature in TBST with 1% nonfat milk. After three additional washes with TBST, the membrane was developed with ECL reagent. The images of the Rubisco large subunit (55 kDa) band were collected with a ChemDoc XRS+ system (Bio-Rad, USA).

## Apparent photo conversion efficiency calculations

The apparent photo conversion efficiency (APCE) values in this work were calculated according to Wagner et al.[41]. We define it as the conversion efficiency of the actual incident light to biomass:

$$APCE(\%) = E_B/E_I \tag{1}$$

$E_B$ – energy fixed in biomass (J); $E_I$ – energy in actual incident light (J).

For the calculation of $E_B$, the calorific value ($H_B$) of 23.4 KJ g$^{-1}$ for *C. pyrenoidosa*[42] was taken to calculate the chemical energy stored in

biomass. During two days cultivation, $E_B$ can be calculated as:

$$E_B = \upsilon_B \times H_B \times V \times t \qquad (2)$$

$\upsilon_B$—biomass growth rate (g L$^{-1}$ day$^{-1}$); $H_B$—calorific value of biomass (J g$^{-1}$), here assumed to be 23.4 KJ g$^{-1}$; V—working volume of photobioreactor, 20 mL; t—process time, 2 days.

For the calculation of $E_I$, a 7IGF10 grating spectrograph (Saifan Optoelectronic Instrument Co., Ltd., China) was used to measure the spectrum of light source. Photosynthetically active radiation (PAR) range of 400–700 nm is considered for the calculation of the average quantum energy of photons (U):

$$U = \int_{400nm}^{700nm} \frac{\phi(\lambda)hc}{\lambda} d\lambda \div \int_{400nm}^{700nm} \phi(\lambda)d\lambda \qquad (3)$$

U – average quantum energy of photons (J); $\phi(\lambda)$ – photon flux at different wavelength (count per second, arbitrary unit); h – Planck's constant ($6.626 \times 10^{-34}$ J s); c – speed of light ($2.998 \times 10^8$ m s$^{-1}$); $\lambda$ – wavelength (nm); A QSL-2101 radiometers (Biospherical Instruments Inc., USA) with an integrating sphere was used to measure the actual incident light intensity (reflected and diffused light by photobioreactor and medium was excluded) in a photobioreactor filled with fresh BG-11 medium.

The total energy in actual incident light $E_I$ is:

$$E_I = N_A \times U \times I \times A \times t \qquad (4)$$

$E_I$—energy in actual incident light (J); $N_A$—Avogadro's constant ($6.022 \times 10^{23}$ mol$^{-1}$); U—average quantum energy of photons (Joule per photon = J); I—actual incident light intensity, measured to be 48.8 μmol m$^{-2}$ s$^{-1}$; A—cross section area of algal suspension, measured to be 12 cm$^2$; t—process time, 2 days; Values were averaged over three independent experiments. As a result, the apparent photo conversion efficiency (APCE) values are 5.1% for the control group (bare *C. pyrenoidosa*) and 9.8% for MOF/Algae hybrids.

### Protein content measurements

*C. pyrenoidosa* and *C. pyrenoidosa*/MOF were harvested by centrifugation (2220 g, 5 min) for three times, then resuspended in protein extraction buffer for ultrasonic treatment for cell disruption. Then the homogenate was centrifugated at 12000 g for 10 min at 4 °C. For the determination of extracellular protein in medium, the supernatants of algal suspensions were directly centrifuged (12000 g, 5 min) and concentrated 10-fold by N$_2$ purging treatment. The protein content in supernatant was determined using BCA protein assay kit (Meilunebio, China). A calibration curve was established in advance using bovine serum albumin (BSA) gradient concentration solutions by the same method.

### Carbohydrate content measurements

*C. pyrenoidosa* and *C. pyrenoidosa*/MOF were washed and collected by centrifugation (2220 g, 5 min) for three times, then freeze dried and milled to powder. 5 mg algal powder was dispersed and disrupted in 2 mL deionized water by sonication, and adjusted the volume to 5 mL. In total 5 mL anthrone test solution (1 g L$^{-1}$ anthrone/80% H$_2$SO$_4$) and 0.2 mL of the homogenate were added into 0.8 mL deionized water, the mixture was boiled in boiling water for 10 min. Subsequently, the absorbance of the mixture at 625 nm was detected, and the carbohydrate concentration was calculated according to a calibration curve mapped with glucose standard solutions using the same method.

### Lipid content measurements

A total of 50 mg freeze-dried algal powder of *C. pyrenoidosa* and *C. pyrenoidosa*/MOF was weighted and put into a 5 mL plastic centrifuge tube, using 2 mL lipid extraction solvent of Chloroform/methanol (2:1) extracted for 30 min and repeated three times. The extraction supernatant was collected and combined, then completely evaporate the solvent in a nitrogen blowing concentrator. The content of lipid can be weighed and calculated.

### Statistical analysis

Unless otherwise indicated, statistical analysis was performed in OriginPro version 9. The samples chosen for analysis were derived from at least three biologically independent experiments. Data are presented as graphs or in-text showing the mean values $\pm$ SD as appropriate.

### Reporting summary

Further information on research design is available in the Nature Portfolio Reporting Summary linked to this article.

## Data availability

The authors declare that data supporting the findings of this study are available within the article and its Supplementary Information files. Source data are provided in this paper.

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

## Acknowledgements

This work was financially supported by the National Natural Science Foundation of China for "Fundamental Research Center on Artificial Photosynthesis (FReCAP)" (22088102), the National Natural Science Foundation of China (22075280), the National Key R&D Program of China (No. 2019YFA0904600), the National Natural Science Foundation of China (No. 22102178). W.W. thanks the support from Youth Innovation Promotion Association of Chinese Academy of Sciences (2020191). We thank H. X. Han for English polishment and Energy Biotechnology Platform of Dalian Institute of Chemical Physics for providing instrumental support.

## Author contributions

C.L. conceived the idea and proposed the project. D.L. did all the bio-related experiments and wrote the manuscript. H.D. prepared and characterized MOF materials. C.L. and W.W. discussed the results and revised the manuscript. X.C. participated in the discussion of experimental results.

## Competing interests

The authors declare no competing interests.
