## [Peer Review File · Nature Communications]

REVIEWER COMMENTS

Reviewer #1 (Remarks to the Author):

In this paper, Li and co-authors report on the enhancement of photosynthesis in the microalga *Chlorella* by using a metal organic framework (MOF) which self-assembles at the surface of algae. This paper reports interesting observations related to a stimulating effect of MOFs on biomass production, but would need much better characterization of the stimulating effect to conclude to a stimulating effect on algal photosynthesis.

The absorbance spectra of the *Chlorella* suspension are rather strange, the chlorophyll peak is not well-resolved and the general trend is an increase of absorbance at higher wavelengths, which is not in line with what is generally observed with algal suspensions where a decrease of absorbance at higher wavelength is observed. This may indicate the presence of diffusive materials (like algal debris) in algal samples. A better characterization of algal suspensions in the presence of MOFs should be performed, for instance by using flow cytometry, in order to determine the number and size of cells and free particles present in the suspensions.

Based on absorbance spectra, the authors argue that MOFs protect PSII from photodamage by UV and do not impair harvesting properties (lines 84-85). This sole experiment is clearly not sufficient to support such a statement. Additional characterization of photosynthetic properties should be carried out to determine changes in light harvesting properties, such as measuring light saturation curves of photosynthetic O₂ production of algal samples and algal-MOF samples of similar cell densities. Moreover, a possible UV protection by MOFs should be demonstrated by using an experimental protocol in which algal samples are subjected to UV stress.

The effect of MOFs on algal growth is then investigated by following OD change at 750 nm. Based on the calibration curve shown on Fig. S9, it is concluded that MOFs induce a two-fold increase in biomass production (Fig. 3a). Clearly, measuring OD change is not sufficient to conclude to changes in biomass production. Indeed, from Fig. 2 b,c it seems highly probable that the diffusive properties of algal cells are changed upon absorption of MOFs. However, the calibration curve shown on Fig. S9 was only made on algal cells without MOFs, and absorption spectra shown on Fig. S5 show similar absorbance at 750 nm, which is rather surprising. What were cell densities in the experiments shown on Fig. S5 and how were the spectra normalized? Clearly, more direct determination of algal biomass must be performed in order to validate the effect of MOFs. Further, it is concluded that MOFs enhance the photosynthetic activity of algae. Again, in order to conclude that photosynthesis is enhanced in the presence of MOFs it is necessary to perform more direct measurements of photosynthesis by measuring either CO₂ uptake or O₂ production, the latter being generally used to characterize the photosynthetic activity of algal suspensions.

At a 50 ppm MOF concentration how many MOFs particles or molecules are present in the solution and how many are available for each single algal cell? Are the MOFs present in large excess? How many MOFs molecule/particle are available per cell when the biomass increases 8-fold as shown on Fig. 3 (at pH7). Again, the determination of the cell concentration is needed here, particularly at different time points during the growth of cells in the presence of MOFs.

The authors observe small particles at the surface of MOFs by TEM and propose that free enzymes diffusing out of the cell are adsorbed on MOFs. Then, based on the effect of the carbonic anhydrase inhibitor acetazolamide and on the measurement of carbonic anhydrase activity in the supernatant, it is concluded that the external carbonic anhydrase adsorbed on MOFs would supply bicarbonate to algal, the conversion of CO₂ into HCO₃⁻ being catalyzed by the adsorbed carbonic anhydrase. Such a scenario should be supported by more direct evidence of the presence of carbonic anhydrase on the surface of MOFs (e.g. western blot, immunolocalization...). Further, the fact that the MOF-induced increase in biomass is similar at pH6 and pH8 (Fig. 3b), conditions in which HCO₃⁻ concentration at

the equilibrium with air are extremely different, is difficult to explain in the framework of the proposed scenario.

More minor concerns:

- I do not quite well understand what information is expected from the pH drift measurements (Fig. 3c) since the authors use a BG11 medium buffered with 20mM HEPES
- Many experimental details are missing, particularly in supplemental Figures.
- Rubisco determination performed on Fig. S14 is not satisfactory. A western blot should be performed to ensure the shown band is due to Rubisco. Also, the total amount of proteins loaded on the gel should be indicated.

Reviewer #2 (Remarks to the Author):

This is a very novel study on the environmental effects of MOF materials. It reported for the first time that MOF could adsorb CO₂ and transport it to microalgae, which resulted in higher photosynthetic efficiency. The phenomenon is interesting and provides a new approach to concentrate and transport atmospheric CO₂ towards to the Rubisco carboxylation. However, there are several issues to address.

1. The chlorophyll contents of *C. pyrenoidosa* in the presence/absence of MOFs should be measured.
2. Is there any possibility to measure the HCO₃⁻ contents in solution and in *C. pyrenoidosa*? HCO₃⁻ content change is the key issue of this study.
3. If MOF enhanced the CO₂ adsorption and generate more HCO₃⁻, the pH should be lower. Why Alga/MOF groups have higher pH than Alga groups in Fig. 3.
4. So many MOF particles were found on the alga surface in Fig. 2. Would they damage the cell wall/membrane? Why MOF particles were not found in TEM observation in Fig. S16?
5. Fig. S10 does not have growth curves. Please check the figure or the caption.
6. The text should be checked and improved. For example, "Here in" should be "Herein" and "shoule" should be "should".

Reviewer #3 (Remarks to the Author):

Authors found that self-assembly of metal-organic frameworks (MOFs) on one of microalgae can enhance its photosynthetic efficiency by about 2 folds by the conversion of the captured CO₂ to the transportable HCO₃⁻ species at bio-organic interface. This work provided evidence for an artificial approach of concentrating atmospheric CO₂ for enhancing biomass yield of photosynthesis in microalgae. The work is novel and is of significance to the photosynthesis research for improvement of photosynthesis in microalgae although the evidence to support the conclusion is a little weak at present stage. Suggestion and comments are as follows.

Title

The title is too broad because the work focused on a green alga.

Abstract

What kind of microalgae did you use for this work?

Introduction

Given that the research focused on a green alga, it is better to introduce the CCM background of green alga.

Last paragraph: Please provide the full name of *C. pyrenoidosa* when first used the name.

Results

Line 94: "It indicates that the free enzymes are diffused out of the cell and adsorbed on the MOF". What kinds of enzymes are? The sentence is better to be removed to the explanation of Figure 4. Fig.S14. I suggest that authors identify the protein by immunoblot analysis included the loading control.

Discussion

This part is only data analysis. It should be included more previous results for discussion of the other possibilities or related mechanism.

I think that it is important that authors should check whether there is similar effect of MOF treatment on other species of microalgae.

Point-by-point response to the reviewers' comments

We thank the three reviewers for your efforts in reviewing our manuscript and providing constructive comments. All comments and suggested changes were carefully considered and addressed. All changes have been highlighted with yellow color in the revised version of the manuscript.

Reviewer #1

In this paper, Li and co-authors report on the enhancement of photosynthesis in the microalga *C. pyrenoidosa* by using a metal organic framework (MOF) which self-assembles at the surface of algae. This paper reports interesting observations related to a stimulating effect of MOFs on biomass production, but would need much better characterization of the stimulating effect to conclude to a stimulating effect on algal photosynthesis.

Reply:

Thank you for your positive comments and professional suggestions on our work.

Comment 1

The absorbance spectra of the *Chlorella* suspension are rather strange, the chlorophyll peak is not well-resolved and the general trend is an increase of absorbance at higher wavelengths, which is not in line with what is generally observed with algal suspensions

where a decrease of absorbance at higher wavelength is observed. This may indicate the presence of diffusive materials (like algal debris) in algal samples.

Reply:

Thank you for your kind advices. We have carefully checked the absorbance spectra of the *C. pyrenoidosa* suspension and found that the unnormal results were caused by an instrumental error and cuvette pollution. After eliminating the instrumental faults, we obtained reliable results of the UV-Vis spectra. Figure R1 shows that UV-Vis absorption spectra of the MOF/Algae hybrid system. The chlorophyll peak is characterized at the wavelength of about 680 nm. After self-assembly, the absorption spectrum of the hybrid is composed of the spectra both MOFs and the *C. pyrenoidosa*. We have updated the figure in the revised manuscript.

Figure R1 (Revised version of Supplementary Fig. 6). UV-Vis absorption spectra of the hybrids of MOF, *C. pyrenoidosa* and MOF/Algae. OD₇₅₀ of *C. pyrenoidosa*, 0.32; MOF amount, 50 ppm.

Comment 2

A better characterization of algal suspensions in the presence of MOFs should be

performed, for instance by using flow cytometry, in order to determine the number and size of cells and free particles present in the suspensions.

Reply:

Thank you for your advice. Size distribution of algal and MOF/Algae cells are measured by flow cytometry. Figure R2 shows flow cytometric scatter plot and FSC-A (forward scattering-area) distribution of the MOF/Algae system. The size distribution of algal cell in the suspension is uniform (Figure R2a). After self-assembly of algae with MOFs, we find that the main size distribution and number of MOF/Algae cells are the same as that of the bare algal cells (Figure R2b). It's noted that the size of single MOF particle is about 300 nm (Figure 2a). Therefore, self-assembly of MOFs on the algal surface did not change the size of the cell and make them aggregated obviously in the suspension. The results indicate that MOFs in algal suspension efficiently bind on the surface of *C. pyrenoidosa* cell.

We have revised the related contents as follows:

“Algal suspensions in the presence of MOFs are characterized by using flow cytometry to determine the number and size of cells (Supplementary Fig. 5). The size distribution of algal cell in the suspension is uniform (Supplementary Fig. 5a). After self-assembly of algae with MOFs, we find that the main size distribution and number of MOF/Algae cells are the same as that of the bare algal cells (Supplementary Fig. 5b). Therefore, self-assembly of MOFs on the algal surface can not change the size of the cell and make them aggregated in the suspension. The results indicate that MOFs in algal suspension bind on the surface of *C. pyrenoidosa* cell.”

Figure R2 (Revised version of Supplementary Fig. 5). Flow cytometric scatter plot and FSC-A (forward scattering-area) distribution of MOFs (a), *C. pyrenoidosa* in the absence (b) and in the presence (c) of MOFs. Cell density of *C. pyrenoidosa*, 0.2×10^7 cell ml⁻¹.

Comment 3

Based on absorbance spectra, the authors argue that MOFs protect PSII from photodamage by UV and do not impair harvesting properties (lines 84-85). This sole experiment is clearly not sufficient to support such a statement. Additional characterization of photosynthetic properties should be carried out to determine changes in light harvesting properties, such as measuring light saturation curves of

photosynthetic O₂ production of algal samples and algal-MOF samples of similar cell densities. Moreover, a possible UV protection by MOFs should be demonstrated by using an experimental protocol in which algal samples are subjected to UV stress.

Reply:

We have carried out the experiments of photosynthetic O₂ production of the algal samples under saturation light irradiation. As shown in Figure R3, the O₂ evolution rate of the bare algal sample decreases gradually, while the rate of the MOF/Algae sample keeps constant. About of the effect on UV stress, We measured the intrinsic electron transfer rate (ETR) of photosystem II (PSII) by a chlorophyll fluorimeter to characterize the photoinhibition or photodamage of PSII. Figure R4 shows that the effect of UV stress on the ETR of PSII in the algae and MOF/Algae samples versus light intensities. Before irradiation with UV light, the ETR of PSII in the algae is the same as that the MOF/Algae sample. But after 30 min 10 $\mu\text{E m}^{-2} \text{s}^{-1}$ UV light irradiation, they show the difference with light intensity from 750 $\mu\text{E m}^{-2} \text{s}^{-1}$ to 2500 $\mu\text{E m}^{-2} \text{s}^{-1}$, and the ETR of PSII in the MOF/Algae sample performs higher than that in the bare algae. The results indicate the MOFs self-assembled on the surface of algal cell can block the UV light to a certain extent.

Thus, summarize comments 1 and 3, we have revised the related contents as follows:

“UV-Vis absorption spectra of the MOF/Algae hybrid system is characterized (Supplementary Fig. 6). After self-assembly, the absorption spectrum of the hybrid is composed of both MOFs and bare alga. The light shade of MOFs on the cell surface at the wavelength of less than 400 nm is in favor of protecting photosystem II against UV

light. To investigate the photoprotection of photosystem II, we carry out the experiments of photosynthetic O₂ production of the algal samples under saturation light irradiation (Supplementary Fig. 7). The O₂ evolution rate of the bare algal sample decreases gradually, while the rate of the MOF/Algae sample keeps constant. And the intrinsic electron transfer rate (ETR) of photosystem II (PSII) by a chlorophyll fluorimeter to characterize the photoinhibition or photodamage of PSII (Supplementary Fig. 8). Before irradiation with UV light, the ETR of PSII in the algae is the same as that the MOF/Algae sample. But after 30 min 10 μE m⁻² s⁻¹ UV light irradiation, they show the difference with light intensity from 750 μE m⁻² s⁻¹ to 2500 μE m⁻² s⁻¹, and the ETR of PSII in the MOF/Algae sample performs higher than that in the bare algae. These results indicate the MOFs self-assembled on the surface of algal cell can protect PSII against photodamage by the UV light to a certain extent.”

The experimental methods have added in the revised supporting information file.

Figure R3 (Revised version of Supplementary Fig. 7). Kinetics of photosynthetic oxygen evolution and average oxygen evolution rates under 1000 μE m⁻² s⁻¹ irradiation

($\lambda > 600$ nm) of *C. pyrenoidosa* in the absence and in the presence of MOF. Error bars represent the standard deviation of three experimental results.

Figure R4 (Revised version of Supplementary Fig. 8). The plots of electron transfer rates of PSII versus light intensities of *C. pyrenoidosa* in the absence and in the presence of MOF before and after 30 min $10 \mu\text{E m}^{-2} \text{s}^{-1}$ UV irradiation ($\lambda = 254$ nm). Error bars represent the standard deviation of three experimental results.

Comment 3

The effect of MOFs on algal growth is then investigated by following OD change at 750 nm. Based on the calibration curve shown on Fig. S9, it is concluded that MOFs induce a two-fold increase in biomass production (Fig. 3a). Clearly, measuring OD change is not sufficient to conclude to changes in biomass production. Indeed, from Fig. 2 b,c it seems highly probable that the diffusive properties of algal cells are changed upon absorption of MOFs. However, the calibration curve shown on Fig. S9 was only made on algal cells without MOFs, and absorption spectra shown on Fig. S5 show similar absorbance at 750 nm, which is rather surprising. What were cell densities in

the experiments shown on Fig. S5 and how were the spectra normalized? Clearly, more direct determination of algal biomass must be performed in order to validate the effect of MOFs.

Reply:

It's conformed that the MOFs boost the biomass prudction. Indeed, using OD change to quantify the amount of biomass is inaccurate or imprecise in the presence of MOFs. Due to the difference of absorbance spectra at 750 nm, there is an effect of MOFs on the amount of dry cell weight with the OD change. According your suggestion, we have updated the calibration curve of MOF/Algae samples. Figure R5 shows the calibration curve of the dry cell weight and algal cell number versus the optical density at 750 nm of *C. pyrenoidosa* in the absence and in the presence of MOF. The cell numbers per OD₇₅₀ that is equal to 35.5 in the presence of MOFs are less than that ratio of 40.9 in the absence of MOF in 0.04 mm² square. Therefore, based on the accurate relationship, we calculate that the average biomass growth rate of MOF/Algae is 0.25 g L⁻¹ day⁻¹, which is slightly less than the quantity of the algae alone. Therefore, the biomass growth rate of MOF/Algae is about 1.9 times as that of the bare algae.

We have added the calibration curves as Figure S12 in the supproting information and reivsed all the data related the biomass amount and the photosynthetic efficiency in the revised manuscript.

Figure R5 (Supplementary Fig.12). The calibration curves of (a) algal cell number (on 0.2×0.2 mm² square) versus the optical density at 750 nm (OD₇₅₀) of *C. pyrenoidosa* in the absence and in the presence of MOF, and (b) the dry cell weight (DCW, net biomass without water). Error bars represent the standard deviation of three experimental results.

Comment 4

Further, it is concluded that MOFs enhance the photosynthetic activity of algae. Again, in order to conclude that photosynthesis is enhanced in the presence of MOFs it is necessary to perform more direct measurements of photosynthesis by measuring either CO₂ uptake or O₂ production, the latter being generally used to characterize the photosynthetic activity of algal suspensions.

Reply:

We agree that O₂ evolution activity is often used for researching photosynthesis. O₂ production of the MOF/Algae system is measured by Clark-type oxygen electrode. Figure R6 shows kinetic plots of photosynthetic oxygen evolution and average oxygen evolution rates of *C. pyrenoidosa* in the absence and in the presence of MOF. The O₂

evolution amount of the MOF/Algae system increased by 14.3% compared with the bare algae sample within the monitor time of 10 minutes. Although the light source is equipped with a cutoff wavelength filter ($\lambda > 600$ nm) and the flux of $50 \mu\text{E m}^{-2} \text{s}^{-1}$ irradiation, the rate of the bare algae sample is decreasing slightly. The result indicates that CO_2 supply is not enough for carbon fixation by utilizing the reducing equivalents generated from the light reaction of water oxidation utilizing solar light. The photosynthetic efficiency depends on coupling of the light reaction and the dark reaction. For long time culture such as two days like the condition of figure 3a and 3b, the difference is more significant because of an ample supply of CO_2 .

Figure R6. Kinetics of photosynthetic oxygen evolution and average oxygen evolution rates of *C. pyrenoidosa* in the absence and in the presence of MOF. Cell density of 0.5×10^7 cell ml^{-1} and the light source with a cutoff wavelength filter (under $50 \mu\text{E m}^{-2} \text{s}^{-1}$ irradiation, $\lambda > 600$ nm). Error bars represent the standard deviation of three experimental results.

Comment 4

At a 50 ppm MOF concentration how many MOFs particles or molecules are present in the solution and how many are available for each single algal cell? Are the MOFs present in large excess? How many MOFs molecule/particle are available per cell when the biomass increases 8-fold as shown on Fig. 3 (at pH 7). Again, the determination of the cell concentration is needed here, particularly at different time points during the growth of cells in the presence of MOFs.

Reply:

There are about 4.1×10^{12} MOF particles available for single algal cell in 1 L algal suspension at 50 ppm MOF concentration calculated based on the single particle with average size of 300 nm and octahedral morphology. And the MOFs are large excess compared with the number of algal cell in the solution during the culturing time. The ratio of MOF particles per cell is about 400 when the biomass increases 8-fold as shown on Fig. 3 (at pH 7). Even the binding efficiency of MOFs to algal cell is not 100%, MOFs should present in large excess even at the latter cultivation period of *C. pyrenoidosa*.

About cell concentration, according to Figure R5, we have calculated the relationship of cell concentration with dry weight of cell based on the OD value. The cell density of *C. pyrenoidosa* is calibrated to be 6.3×10^{10} cells per gram DCW for bare algae, and 6.0×10^{10} cells per gram DCW for MOF/Algae. The parameters have been added in the legend of Figure 3 and main text of the revised manuscript.

Comment 5

The authors observe small particles at the surface of MOFs by TEM and propose that free enzymes diffusing out of the cell are adsorbed on MOFs. Then, based on the effect of the carbonic anhydrase inhibitor acetazolamide and on the measurement of carbonic anhydrase activity in the supernatant, it is concluded that the external carbonic anhydrase adsorbed on MOFs would supply bicarbonate to algal, the conversion of CO₂ into HCO₃⁻ being catalyzed by the adsorbed carbonic anhydrase. Such a scenario should be supported by more direct evidence of the presence of carbonic anhydrase on the surface of MOFs (e.g. western blot, immunolocalization...).

Reply:

According to your suggestion, we have tried an antibody recognizing Carbonic anhydrase II rat monoclonal antibody (Catalog No. AG1271, Beyotime, China), but no band was detected. It's difficult to find the available antibody for algae-derived CAs. Carbonic anhydrase is a kind of important enzymes that catalyze reversible conversion between CO₂ and HCO₃⁻ through hydration and dehydration. There are various carbonic anhydrase enzymes in the photosynthetic cells, which play the critical roles in the overall processes from CO₂ capturing, transporting and concentrating to final fixation by Rubisco enzyme. Determination of enzymatic activity is a simple and valid method to confirm the carbonic anhydrase in the photosynthesis research (The Plant Journal (2015) 82, 429–448; Annu. Rev. Plant Biol. 2020. 71:18.1–18.25). The carbonic anhydrase located on the cell surface of *C. pyrenoidosa* had been found in both the soluble and insoluble types for CO₂ utilization (Plant & Cell Physiol. 24(3): 441-451 (1983)). In our work, we verify external carbonic anhydrase (eCA) and its activity by

conducting inhibition experiments (Fig. 4b) and testing CO₂ hydration with water into HCO₃⁻ and protons (Fig. 4c and d). An important result is that the CO₂ hydration rate of eCA in the presence of MOF is higher than that of free eCA in the supernatant. Based on the results and TEM imagings, we depict the proposed scenario of CO₂ capture and concentrating.

Comments 6

Further, the fact that the MOF-induced increase in biomass is similar at pH 6 and pH 8 (Fig. 3b), conditions in which HCO₃⁻ concentration at the equilibrium with air are extremely different, is difficult to explain in the framework of the proposed scenario.

Reply:

Normally, we agree with you on that the HCO₃⁻ concentrations at the equilibrium are different at pH 6 and pH 8 in aqueous solution. Since pK_{a1} of carbonic acid is about 6.4, CO₂ is the major form at pH 6 and HCO₃⁻ at pH 8 in algal suspension respectively. However, in our case that MOFs self-assembled on the surface of cell, the proposed scenario is two sequential processes of CO₂ concentraing at the bio-inorganic interface: (1) the capture of CO₂ by MOF, and (2) the conversion of CO₂ to HCO₃⁻ by extracellular carbonic anhydrase and OH⁻. At pH 6, the former is fast while the latter is slow due to acidic environment. While at pH 8, the former is slow due to the lack of CO₂ to be captured by MOF while the latter is fast. Therefore, their biomass increases are similar.

More minor concerns:

Comment 6

I do not quite well understand what information is expected from the pH drift measurements (Fig. 3c) since the authors use a BG11 medium buffered with 20mM HEPES

Reply:

The value of pH drift indicates the nitrogen assimilation rate of *C. pyrenoidosa* which produces hydroxide when using nitrate as inorganic nitrogen source (Annu. Rev. Plant Biol. 56(1): 99-131 (2005)). Since the inorganic nitrogen source is NO_3^- , OH^- is inevitably accumulated during the assimilation of NO_3^- to amino acids, which can lead to pH value up to 10 and hampers algal growth. The chemical equation which reflects the conversion of major elements during the cultivation of *C. pyrenoidosa* in BG11 medium is listed below. Typical cultivation of *C. pyrenoidosa* in photobioreactor solves the problem by elevating CO_2 concentration (e.g. 2%) in airflow to resist the OH^- accumulation. In our system, atmospheric CO_2 can't stop the pH drift to high pH, so the addition of pH buffer solution is necessary to maintain the normal cultivation. Meanwhile, the pH drift rate can also indicate the nitrogen assimilation rate of alga, as well as the biomass accumulation rate.

Comment 7

Many experimental details are missing, particularly in supplemental Figures.

Reply:

Thank you for your kind advice. We have carefully added the details in the revised manuscript and supporting information files with yellow highlight.

Comment 8

Rubisco determination performed on Fig. S14 is not satisfactory. A western blot should be performed to ensure the shown band is due to Rubisco. Also, the total amount of proteins loaded on the gel should be indicated.

Reply:

Thank you for your kind advices. We identified the protein by immunoblot with an anti-Rubisco monoclonal antibody (Catalog No. AG5359, Beyotime, China). A specific band was detected at the expected molecular weight corresponding to the large subunit of Rubisco (Figure R7). Therefore, we confirmed that the band detected in Fig. S14 is Rubisco. We have added the figure R7 as Supplementary Fig. 20(b) in the revised supplementary information.

Figure R7 (Revised version of Supplementary Fig. 20). Western blot for Rubisco in

the homogenate of *C. pyrenoidosa*. The total protein amount, 1.73 μg ; Cultivation conditions: temperature, 26°C; light intensity, 50 $\mu\text{E m}^{-2} \text{s}^{-1}$; 20 mL min^{-1} ambient airflow; 24 hours continuous illumination. Three numbers represent three biological replica.

Reviewer #2 (Remarks to the Author):

This is a very novel study on the environmental effects of MOF materials. It reported for the first time that MOF could adsorb CO_2 and transport it to microalgae, which resulted in higher photosynthetic efficiency. The phenomenon is interesting and provides a new approach to concentrate and transport atmospheric CO_2 towards to the Rubisco carboxylation. However, there are several issues to address.

Thank you for your positive comments.

Comment 1

The chlorophyll contents of *C. pyrenoidosa* in the presence/absence of MOFs should be measured.

Reply:

The chlorophyll contents of *C. pyrenoidosa* are measured. Figure R8 shows the contents of photosynthetic pigments of *C. pyrenoidosa* in the presence/absence of MOFs. The contents of chlorophyll a, chlorophyll b and carotenoids of *C. pyrenoidosa* are the same whether in the presence of MOFs or not. It indicates that biomass increasing has no effect on the amount of photosystem II and photosystem I proteins.

We have added the results in the revised manuscript as follow:

“The contents of chlorophyll a, chlorophyll b and carotenoids of *C. pyrenoidosa* are the same whether in the presence of MOFs or not. Biomass increasing has no effect on the amount of photosystem II and photosystem I proteins (Supplementary Fig. 14)”

Figure R8 (Revised version of Supplementary Fig. 14). The contents of photosynthetic pigments (Chl a, chlorophyll a; Chl b, chlorophyll b; Cx, carotenoids) of *C. pyrenoidosa* in the absence and in the presence of MOF after two-day cultivation and extracted by anhydrous ethanol. Error bars represent the standard deviation of three experimental results.

Comment 2

Is there any possibility to measure the HCO_3^- contents in solution and in *C. pyrenoidosa*?

HCO_3^- content change is the key issue of this study.

Reply:

We agree with you that HCO_3^- as the main form of CO_2 is crucial to understand the mechanism of CO_2 fixation in the work. The HCO_3^- contents in solution and in *C.*

pyrenoidosa is possible to be measured by Raman spectroscopy. And we tried to use the method to measure the content change of the HCO_3^- , but it's difficult to obtain viable data under the rather low concentration in solution and especially in *C. pyrenoidosa*. More sensitive detection method need to be developed for further use. Therefore, We have measured the contents of dissolved inorganic carbon (DIC, including CO_2 , HCO_3^- and CO_3^{2-}). Figure R9 shows the DIC in the media of MOF/Algae and bare algae after two-day cultivation. DIC in medium of the MOF/Algae system is slight higher than that of the bare with the initial pH from 6 to 8. The DIC increase is derived from the MOFs capturing CO_2 .

We have added the results in the revised manuscript as follow:

“Meanwhile, it's noted that DIC in medium of the MOF/Algae system is slightly higher than that of bare algae with the initial pH from 6 to 8. The DIC increase is derived from the MOFs capturing CO_2 (Supplementary Fig. 19)”

Figure R9 (Supplementary Fig. 19). The contents of dissolved inorganic carbon (DIC, including CO_2 , HCO_3^- and CO_3^{2-}) in medium of *C. pyrenoidosa* in the absence and presence of MOF under different initial pH value before (Blank, only medium without

algal cells) and after two-day cultivation (Algae and MOF/Algae). Error bars represent the standard deviation of three experimental results.

Comment 3

If MOF enhanced the CO₂ adsorption and generate more HCO₃⁻, the pH should be lower. Why Alga/MOF groups have higher pH than Alga groups in Fig. 3.

Reply:

The value of pH drift in Fig.3c is caused by the nitrogen assimilation of *C. pyrenoidosa* which produces hydroxide when using nitrate as inorganic nitrogen source. The chemical equation which reflects the conversion of major elements during the cultivation of *C. pyrenoidosa* in BG11 medium is listed below. Since the inorganic nitrogen source is NO₃⁻, OH⁻ is inevitably accumulated during the assimilation of NO₃⁻ to synthesize amino acids. The pH drift rate can also indicate the biomass accumulation rate. Because MOF/Algae grows faster than bare algae lead the more OH⁻ accumulation, a higher pH of MOF/Algae groups than that of algae groups.

Comment 4

So many MOF particles were found on the alga surface in Fig. 2. Would they damage the cell wall/membrane? Why MOF particles were not found in TEM observation in Fig. S16?

Reply:

We observe that MOF particles are assembled on the surface of *C. pyrenoidosa* (Figure 2c and 2d). The opposite electrostatic potentials between MOF and *C. pyrenoidosa* tend to drive their self-assembly by van der Waals interaction (Figure 1d). Due to the abundant carboxyl group on the surface of MOFs, it is biocompatible for binding with the cell wall through weak interaction without damage. In addition, algal suspensions in the presence of MOFs are characterized by using flow cytometry and we find that self-assembly of MOFs on the algal surface can not change the size of the cell and make them aggregated in the suspension (Revised version of Supplementary Fig.5). The sample is the bare *C. pyrenoidosa* in Fig. S16, which is used for observing the intrinsic cell structure of *C. pyrenoidosa*.

Comment 5

Fig. S10 does not have growth curves. Please check the figure or the caption.

Reply:

We must apologize for the mistake. We have corrected the figure and the caption of “ligand concentration” in Fig. S10a, and “biomass growth” in Fig. S10b.

Comment 6

The text should be checked and improved. For example, “Here in” should be “Herein” and “shoule” should be “should”.

Reply:

Thank you for your kind advice. We have corrected these spelling mistakes and

examined the sentences in our manuscript carefully.

Reviewer #3 (Remarks to the Author):

Authors found that self-assembly of metal-organic frameworks (MOFs) on one of microalgae can enhance its photosynthetic efficiency by about 2 folds by the conversion of the captured CO₂ to the transportable HCO₃⁻ species at bio-organic interface. This work provided evidence for an artificial approach of concentrating atmospheric CO₂ for enhancing biomass yield of photosynthesis in microalgae. The work is novel and is of significance to the photosynthesis research for improvement of photosynthesis in microalgae although the evidence to support the conclusion is a little weak at present stage. Suggestion and comments are as follows.

Thank you for your positive comments and professional suggestions on our work.

Comment 1

Title

The title is too broad because the work focused on a green alga.

Reply:

Thank you for your kind advice. We have changes the title as “Enhancing photosynthetic efficiency by assembling metal-organic frameworks on *Chlorella pyrenoidosa*”.

Comment 2

Abstract

What kind of microalgae did you use for this work?

Reply:

Thank you for your kind advice. We have added the full name of the microalga *Chlorella pyrenoidosa* in Abstract.

Comment 3

Introduction

Given that the research focused on a green alga, it is better to introduce the CCM background of green alga.

Reply:

Thank you for your kind advice. We have extended the introduction of the intrinsic CCM in green alga. (see page 2, lines 10-26 in the revised manuscript).

Comment 4

Last paragraph: Please provide the full name of *C. pyrenoidosa* when first used the name.

Reply:

Thank you for your kind advice. We have added the full name of *C. pyrenoidosa* as *Chlorella pyrenoidosa*.

Comment 5

Results

Line 94: “It indicates that the free enzymes are diffused out of the cell and adsorbed on the MOF”. What kinds of enzymes are? The sentence is better to be removed to the explanation of Figure 4.

Reply:

The enzyme absorbed on the surface of MOFs is one of kind carbonic anhydrase. We observe the small particles on the MOFs from the SEM data of Fig. 2d and infer that the particles may be a kind of free enzyme. Subsequently, we verify external carbonic anhydrase, eCA and its activity by conducting inhibition experiments (Fig. 4b) and testing CO₂ hydration with water into HCO₃⁻ and protons (Fig. 4c and d), which is a simple and valid method to confirm the carbonic anhydrase in the photosynthesis research (The Plant Journal (2015) 82, 429–448; Annu. Rev. Plant Biol. 2020. 71:18.1–18.25; *Photosynth. Res.* **77**, 83-94 (2003)). The carbonic anhydrase located on the cell surface of *C. pyrenoidosa* had been found in both the soluble and insoluble types for CO₂ utilization (Plant & Cell Physiol. 24(3): 441-451 (1983)). Therefore, these free enzymes are extracellular carbonic anhydrase.

Comment 6

Fig.S14. I suggest that authors identify the protein by immunoblot analysis included the loading control.

Reply:

We identified the protein by immunoblot with an anti-Rubisco monoclonal antibody

(Catalog No. AG5359, Beyotime, China). A specific band was detected at the expected molecular weight corresponding to the large subunit of Rubisco (Figure R11). Therefore, we confirmed that the band detected in Fig. S14 is Rubisco.

Figure R11 (Revised version of Supplementary Fig. 20). Western blot for Rubisco in the homogenate of *C. pyrenoidosa*. The total protein amount, 1.73 μg ; Cultivation conditions: temperature, 26°C; light intensity, 50 $\mu\text{E m}^{-2} \text{s}^{-1}$; 20 mL min^{-1} ambient air flow; 24 hours continuous illumination. Three numbers represent three biological replica.

Comment 7

Discussion

This part is only data analysis. It should be included more previous results for discussion of the other possibilities or related mechanism.

Reply:

We have rewritten this part as you suggested in the new version of the manuscript as follows:

“Engineering biophysical CCM into agricultural plants such as rice and wheat is an attractive to enhance yields for solving global food demand (*Annu. Rev. Plant Biol.* **71**, 461-485 (2020); *Proc. Natl. Acad. Sci.* **108**, 20260-20264 (2011)). The strategies are to recombine the exogenous CCM module into the chassis plant by the tools of synthetic biology. Compared to CCM in C₄ plants, CCMs in cyanobacteria and algae which rely on CO₂ conversion and transportation are more likely to be implanted into primary crops (*Annu. Rev. Plant. Biol. & Plant. Mol. Bio.* **50**, 539-570 (1999)). The structure and functions of the membrane-bound HCO₃⁻ transporters and intracellular carbonic anhydrases which dominate CCM processes are gradually revealed (*Nat. Plants* **5**, 1184-1193 (2019); *Proc. Natl. Acad. Sci.* **118**, e2107425118 (2021)). Cyanobacterial CCM compartment carboxysome can be introduced into tobacco chloroplast by genetic method for improving CO₂ conversion, but the CO₂ fixation rate decreased because of the mismatch among dark reactions (*Nat. Commun.* **9**, 3570 (2018)). It still need to make effort on understanding the mechanism of CO₂ fixation. In our work, the non-genetic approach that capturing and concentrating CO₂ by MOFs is employed as artificial CCM to be coupled with the intrinsic CCM pathways of *C. pyrenoidosa*.”

Comment 8

I think that it is important that authors should check whether there is similar effect of MOF treatment on other species of microalgae.

Reply:

Thank you for your kind advice. We carry out similar experiments on a model green

alga, *Chlamydomonas reinhardtii*. Figure R12 shows the growth curves of *Chlamydomonas reinhardtii* with the culture time. We find that *Chlamydomonas reinhardtii* self-assembled with MOFs give rise to biomass production increase by about 23%. Although the effect of MOFs with *Chlamydomonas reinhardtii* is not as obvious as that with *Chlorella pyrenoidosa* because of the non-optimized culture condition and species difference, it's confirmed that the MOFs with the function of CO₂ capture can boost algal photosynthetic carbon fixation. Further more experiments need to be conducted to understand the effect on other species of green algae.

We have added the related contents in the revised manuscript as follows:

“To verify whether there is similar effect of MOF treatment on other species of microalgae, we carry out similar experiments on a model green alga, *Chlamydomonas reinhardtii*. It's found that *Chlamydomonas reinhardtii* self-assembled with MOFs give rise to biomass production increase by about 23% after culturing for 2 days (Supplementary Fig. 16).”

Figure R12. (Revised version of Supplementary Fig. 16) The growth curves of *Chlamydomonas reinhardtii* with the addition of 50 ppm MOF at initial pH 7 and be

aerated with ambient air. Cultivation conditions: temperature, 26°C; light intensity, 50 $\mu\text{E m}^{-2} \text{s}^{-1}$; 20 mL min^{-1} ambient air flow; 24 hours continuous illumination. Error bars represent the standard deviation of three experimental results.

REVIEWER COMMENTS

Reviewer #1 (Remarks to the Author):

In this modified version the authors have correctly taken into account some of my concerns and significantly improved the manuscript. However, I still think that the characterization of the photosynthetic activity of MOFS is too scarce and is not sufficient to support the proposed mechanisms for improved CO₂ assimilation based on a MOF-dependent CO₂ concentration.

The novel Fig. S7 (and Figure R6) shows a slight increase in the O₂ production rate in algae when in the presence of MOF, which is clearly not sufficient to conclude to an improved photosynthetic activity by surface-bound MOF which would play a role of CO₂ concentrator. Indeed, if the system acts as a CO₂ concentrator, as claimed by the authors, the improvement of photosynthetic activity by MOF should depend on the CO₂ availability, a much stronger effect should be observed under limiting CO₂ than under non-limiting CO₂ concentration. In other words, the apparent affinity of the photosynthetic O₂ production for C_i should be affected as it is the case when algae harboring or not a CCM are compared (for instance when comparing high CO₂ and low CO₂ grown cells). Clearly, the authors should perform experiments at different C_i levels (and different pH see below), as it is generally done in CCM studies.

The authors claim that the function of MOF and this is coupled to that of the intrinsic CCM of *C. pyrenoidosa*. However, the interaction of MOF with the CCM has not been investigated. To conclude so, the authors should compare the effect of MOF on high-CO₂ and low-CO₂ grown *C. pyrenoidosa* or *C. reinhardtii* cells. If MOF really acts as a CO₂ concentrator, they should observe a much stronger effect on algae not harboring a CCM (high CO₂ grown) as compared to algae harboring a CCM (low CO₂ grown cells).

The proposed scenario assumes the eCA when absorbed on MOF would allow a faster conversion of CO₂ absorbed on MOF into HCO₃⁻ and then a faster carbon assimilation. However, the evidence for such a scenario is rather scarce. First the CA activity absorbed on MOF was not experimentally determined, it is only based on the fact that eCA activity decrease in the external medium in the presence of MOF. To explain that the increase in biomass is similar at pH 6 and pH 8 (reply to my comment #6) the authors argue that in the first case MOF would only concentrate CO₂ and that at pH 8 MOF would have a dual function (increase CO₂ and convert it to HCO₃⁻). In order to support such a scenario, the apparent affinity of net O₂ evolution for C_i should be measured by performing measurements at different pH and different C_i concentrations. The effect of the eCA inhibitor AZA would be also probably needed to support the proposed scenario which would predict a stronger effect of AZA on MOF stimulation at pH 8 than at pH 6.

Reviewer #2 (Remarks to the Author):

My concerns are well addressed. It could be published as it is.

Reviewer #3 (Remarks to the Author):

The revised version is much improved. The evidence to support authors' conclusion is more solid. I am satisfied with this revision and have no further suggestions and comments.

Point-by-point response to the reviewers' comments

Reviewer #1 (Remarks to the Author):

In this modified version the authors have correctly taken into account some of my concerns and significantly improved the manuscript. However, I still think that the characterization of the photosynthetic activity of MOFs is too scarce and is not sufficient to support the proposed mechanisms for improved CO₂ assimilation based on a MOF-dependent CO₂ concentration.

Reply:

Thank you for your efforts in reviewing our manuscript and conferring on us professional comments. We have taken your comments seriously and performed all the experiments carefully to provide more evidences for supporting the proposed scenario. All changes have been highlighted with yellow color in the revised version of the manuscript.

Comment 1

The novel Fig. S7 (and Figure R6) shows a slight increase in the O₂ production rate in algae when in the presence of MOF, which is clearly not sufficient to conclude to an improved photosynthetic activity by surface-bound MOF which would play a role of CO₂ concentrator. Indeed, if the system acts as a CO₂ concentrator, as claimed by the

authors, the improvement of photosynthetic activity by MOF should depend on the CO₂ availability, a much stronger effect should be observed under limiting CO₂ than under non-limiting CO₂ concentration. In other words, the apparent affinity of the photosynthetic O₂ production for C_i should be affected as it is the case when algae harboring or not a CCM are compared (for instance when comparing high CO₂ and low CO₂ grown cells). Clearly, the authors should perform experiments at different C_i levels (and different pH see below), as it is generally done in CCM studies.

Reply:

To systematic investigation of CCM, we measured the net photosynthetic oxygen evolution rates at various C_i (in the form of CO₂ and HCO₃⁻, respectively) of *C. pyrenoidosa* and MOF/*C. pyrenoidosa* cells grown under low CO₂ (LC) and high CO₂ (HC) conditions (Figure R1). The plots are fitted by the Michaelis-Menten equation. We use half saturation constant (K_{1/2}) to indicate the apparent C_i affinity.

Firstly, we compared the cases in the presence of MOF or not when cells were grown under low CO₂ (0.04%, LC). Figure R1a shows that the kinetic plots of the photosynthetic O₂ production is dependent on CO₂ concentration. CO₂ affinity of MOF/*C. pyrenoidosa* cells (K_{1/2} = 12 μM) is stronger than that of bare *C. pyrenoidosa* cells (K_{1/2} = 20 μM). The apparent affinity K_{1/2} of CO₂ increased by 68% indicates the synergetic effect of the functional MOF and the intrinsic CCM in *C. pyrenoidosa* cells.

Secondly, the kinetic plots dependence on the concentration of HCO₃⁻ were conducted as a reference of CO₂. HCO₃⁻ affinities of K_{1/2} around 10 μM are similar for *C. pyrenoidosa* and MOF/*C. pyrenoidosa* cells indicating a fully functional CCM of cells

(Figure R1b). We observed that the CO₂ affinity of MOF/*C. pyrenoidosa* cells ($K_{1/2} = 12 \mu\text{M}$) is close to the $K_{1/2}$ of HCO₃⁻ affinity of around 10 μM (Figure R1e, f). It verifies that MOF as a CO₂ concentrator is favorable for accelerating CO₂ hydration into HCO₃⁻ by eCA.

Finally, the kinetic plots are compared in the cases of cells grown under high CO₂ (HC, 2%). It shows much lower affinities for CO₂ ($K_{1/2} = 202 \mu\text{M}$) and HCO₃⁻ ($K_{1/2} = 71 \mu\text{M}$) compared with control samples when grown under low CO₂ (Figure R1c, d). And the presence of MOF particles doesn't remarkably have influence on the C_i affinities. It indicates that MOF can not play the role in concentrating CO₂ for *C. pyrenoidosa* cells grown in high CO₂ because of not harboring the intrinsic CCM.

Figure R1. (a-d) The plot and the Michaelis-Menten fit of the net O₂ evolution rate versus the concentration of (a, c) CO₂ and (b, d) HCO₃⁻ for *C. pyrenoidosa* (Algae) and MOF/*C. pyrenoidosa* (MOF/Algae) cells grown under (a, b) air (LC, 0.04%) and (c, d) 2% CO₂ (HC) at pH 7. (e-f) The Michaelis constant K_{1/2} values as calculated from the Michaelis-Menten fit of the plot of the net O₂ evolution rate versus the concentration of (e) CO₂ and (f) HCO₃⁻ for *C. pyrenoidosa* (Algae) and MOF/*C. pyrenoidosa* (MOF/Algae) cells grown at pH 7 under air (LC, 0.04%) or 2% CO₂ (HC). Cell density

of 0.5×10^7 cell mL⁻¹ and the light source with a cutoff wavelength filter (under 1500 $\mu\text{E m}^{-2} \text{s}^{-1}$ irradiation, $\lambda > 600$ nm). Error bars represent the standard deviation of three experimental results.

Comment 2

The authors claim that the function of MOF and this is coupled to that of the intrinsic CCM of *C. pyrenoidosa*. However, the interaction of MOF with the CCM has not been investigated. To conclude so, the authors should compare the effect of MOF on high-CO₂ and low-CO₂ grown *C. pyrenoidosa* or *C. reinhardtii* cells. If MOF really acts as a CO₂ concentrator, they should observe a much stronger effect on algae not harboring a CCM (high CO₂ grown) as compared to algae harboring a CCM (low CO₂ grown cells).

Reply:

From the results of Figure R1a and c, we observe that the MOF as a CO₂ concentrator coupled with the intrinsic CCM of *C. pyrenoidosa* enable the CO₂ affinity increased by 68%, which is close to the HCO₃⁻ affinities of K_{1/2} around 10 μM . Furthermore, as shown in Figure R2, biomass growth is measured to verify the effect of MOF acting as a CO₂ concentrator on *C. pyrenoidosa* (Algae) and MOF/*C. pyrenoidosa* (MOF/Algae) cells grown at pH 7 under air (LC, 0.04%) or 2% CO₂ (HC). The addition of MOF obviously promotes the biomass growth to 1.9-fold of *C. pyrenoidosa* cells (harboring a CCM) grown at pH 7 under air (LC, 0.04%), but in the case of cells grown for high CO₂ (2%) without harboring a CCM, no role of MOF is played in enhancing the biomass growth.

Figure R2 (Revised version of Supplementary Fig. 23). Biomass growth curves of *C. pyrenoidosa* (Algae) and MOF/*C. pyrenoidosa* (MOF/Algae) cells grown at pH 7 under air (LC, 0.04%) or 2% CO₂ (HC). Cultivation conditions: temperature, 26°C; light intensity, 50 $\mu\text{E m}^{-2} \text{s}^{-1}$; 20 mL min⁻¹ gas flow; 24 hours continuous illumination. Error bars represent the standard deviation of three experimental results.

Comment 3

The proposed scenario assumes the eCA when absorbed on MOF would allow a faster conversion of CO₂ absorbed on MOF into HCO₃⁻ and then a faster carbon assimilation. However, the evidence for such a scenario is rather scarce. First the CA activity absorbed on MOF was not experimentally determined, it is only based on the fact that eCA activity decrease in the external medium in the presence of MOF.

Reply:

The activities of CA enzyme adsorbed on MOF particles and in medium are assayed by

a colorimetric assay kit (Solarbio, China). It is analyzed according to the hydrolysis of p-nitrophenyl acetate (p-NPA) to p-nitrophenol (p-NP) catalyzed by CA. The method has been added in the revised supporting information. The activity of CA in medium is 0.6 mM h^{-1} and the activity of CA enzyme adsorbed on MOF is 0.2 mM h^{-1} (Figure R3). The adsorption amount of CA proteins on the surface of MOFs is 37% of that in medium. Therefore, after the activities normalized by protein amount, the specific activity of CA on MOFs is similar with that in medium, indicating that eCA adsorbed on MOF enables efficient conversion of MOF-captured CO_2 into HCO_3^- (Figure R3b).

Figure R3 (Revised version of Supplementary Fig. 19). (a) Protein amount and the activity of CA in the medium of microalgal suspension (Medium) and adsorbed by MOF particles (MOF/CA). (b) Relative activity of CA in the medium of *C. pyrenoidosa* suspension (Medium) and the MOF particles that adsorbed CA in the supernatant of microalgal suspension (MOF/CA). $\text{OD}_{750} = 1.5$, $\text{m(MOF)} = 1.5 \text{ mg}$. Error bars represent the standard deviation of three experimental results.

Comment 4

To explain that the increase in biomass is similar at pH 6 and pH 8 (reply to my

comment #6) the authors argue that in the first case MOF would only concentrate CO₂ and that at pH 8 MOF would have a dual function (increase CO₂ and convert it to HCO₃⁻). In order to support such a scenario, the apparent affinity of net O₂ evolution for C_i should be measured by performing measurements at different pH and different C_i concentrations.

Reply:

We have measured the apparent affinity of net O₂ evolution for CO₂ concentration grown under low CO₂ at different pH (Figure R4). We observe that the K_{1/2} for CO₂ of *C. pyrenoidosa* cells is 74 μM at pH 6, decreases to 20 μM at pH 7 and increases to 27 μM at pH 8. And under all the conditions from pH 6 to pH 8, the function of MOF enables the K_{1/2} for CO₂ decreased comparatively. On the other hand, the activity of eCA left in medium are assayed at different pH to understand the process of CCM (Figure R5). The activities are nearly the same at pH 7 and 8, but the activity of CA at pH 6 decreased by 42%. Due to the adsorption of CA on the surface of MOFs, the apparent activities of CA left in medium are reduced correspondingly. Based on the results at different pH, we can make the conclusion that medium pH value plays an important role in the affinities of net O₂ evolution for CO₂ and the activities of CA as well as CO₂ solubility. Even the process of cell physiology is regulated by pH value. These evidences support the proposed scenario.

Figure R4 (Revised version of Supplementary Fig. 21). (a-c) The plot and the Michaelis-Menten fit of the net O₂ evolution rate versus the concentration of CO₂ for *C. pyrenoidosa* (Algae) and MOF/*C. pyrenoidosa* (MOF/Algae) cells grown at (a) pH 6, (b) pH 7 and (c) pH 8 under air (LC, 0.04%). (d) The Michaelis constant K_{1/2} values as calculated from the Michaelis-Menten fit of the plot of the net O₂ evolution rate versus the concentration of CO₂ for *C. pyrenoidosa* (Algae) and MOF/*C. pyrenoidosa* (MOF/Algae) cells grown at different pH under air (LC, 0.04%). Cell density of 0.5×10^7 cell mL⁻¹ and the light source with a cutoff wavelength filter (under $1500 \mu\text{E m}^{-2} \text{s}^{-1}$ irradiation, $\lambda > 600$ nm). Error bars represent the standard deviation of three experimental results.

Figure R5 (Revised version of Fig. 4b). CA activities in the media of *C. pyrenoidosa* (Algae) and MOF/*C. pyrenoidosa* (MOF/Algae) cells grown at different pH under air (LC, 0.04%). Cultivation conditions: temperature, 26°C; light intensity, 50 $\mu\text{E m}^{-2} \text{s}^{-1}$; 20 mL min^{-1} gas flow; 24 hours continuous illumination. Error bars represent the standard deviation of three experimental results.

Comment 5

The effect of the eCA inhibitor AZA would be also probably needed to support the proposed scenario which would predict a stronger effect of AZA on MOF stimulation at pH 8 than at pH 6.

Reply:

We have measured the biomass growth of *C. pyrenoidosa* (Algae) and MOF/*C. pyrenoidosa* (MOF/Algae) cells grown at pH 6, 7 and 8 under air in the presence or absence of 1 mM eCA inhibitor AZA (Figure R6). For pH 6, the addition of AZA shows slight effect on the biomass growth of both *C. pyrenoidosa* and MOF/*C. pyrenoidosa* cells. It is unfavorable for hydration of CO_2 into HCO_3^- thermodynamically. By contrast,

for pH 7 and 8, the addition of AZA caused considerable decrease of biomass growth rate of MOF/*C. pyrenoidosa* cells. The results indicate that activity inhibition of CA depends on the pH value which has an influence to catalytic process of CA as well as the solubility of CO₂ forming to HCO₃⁻ under different condition.

Figure R6 (Revised version of Fig. 4a). The average biomass growth rate of *C. pyrenoidosa* (Algae) and MOF/*C. pyrenoidosa* (MOF/Algae) cells grown at pH 6, 7 and 8 under air with or without the addition of 1 mM eCA inhibitor AZA during two-day cultivation. Cultivation conditions: temperature, 26°C; light intensity, 50 $\mu\text{E m}^{-2} \text{s}^{-1}$; 20 mL min⁻¹ gas flow; 24 hours continuous illumination. Error bars represent the standard deviation of three experimental results.

REVIEWER COMMENTS

Reviewer #1 (Remarks to the Author):

In this new version the authors have addressed my concerns by performing additional experiments to better characterize the photosynthetic activity of algae associated or not with MOFs. The manuscript has been significantly improved, but there are still major improvements to be made.

It is shown (Fig. 4C) that the apparent affinity of photosynthesis for C_i is somehow increased by the presence of MOFs when algae are grown in air but not when they are grown in high CO_2 . It is however difficult to evaluate the effect on the figure since the scale of the X-axis of the graph of Fig. S21 does not allow to distinguish lower CO_2 values (the scale should be extended between 0 and 0.5 to better evaluate the effect visually). The way errors bars shown on Fig. 4C and Fig. R1e,f is not explained and in the absence of statistical analysis, it is difficult to be convinced that the difference observed in $K_{1/2}$ is significant. Moreover, these affinity determinations were performed using a light intensity of $1200 \mu\text{mol photons m}^{-2} \text{s}^{-1}$ which is quite far from the light intensity used for the algal cultivation ($50 \mu\text{mol photons m}^{-2} \text{s}^{-1}$). Altogether this makes the evidence quite weak to explain the observed effect on biomass productivity in terms of affinity. I would suggest the authors to discuss these points both in the result and in the discussion sections.

The authors also observe an increase intracellular accumulation of Rubisco in MOFs-treated algae, but do not really explain what could be the origin of such an effect and what could be its consequence in term of biomass productivity. Is there any evidence in the literature that Rubisco can limit photosynthesis under low CO_2 conditions? How is this additional Rubisco organized within the cell, for instance does it contribute to form larger pyrenoids?

It is not clear how the kinetic properties of Rubisco could be modified. Please discuss this point in relation to the current knowledge of Rubisco activase for instance.

The first paragraph of the discussion has no link with the current work. In connection with my concerns above, I would suggest that authors replace this section with a critical discussion of their own data. How can be the increase in biomass explained? What part is due to an increased affinity for CO_2 and what part could be due to other effects like an increase in the Rubisco content, or a change in kinetics properties of Rubisco? How this relates to limitations previously reported in the literature?

Finally, I am very skeptical about the photosynthetic efficiency calculation (reaching values up to 7.1%), which is made by the authors taking into account the intensity of the incident light intensity and not taking into account light reflection, diffusion... Such calculation would need a careful determination of the light actually absorbed by cells (for instance using an integrating sphere) and cannot be only based on the incident light since light reflection at the surface of the cylinder and the light diffusion have significant contributions. Also, the nature of the light source should be taken into consideration. What is the spectrum of the light source used and what fractions of photons are absorbed by chlorophylls. See for instance Wagner et al. 2016 Biotechnol. J (<https://doi.org/10.1002/biot.201500357>) for the determination of photosynthetic conversion efficiency. Unless the authors would perform a more careful determination of the photosynthetic conversion efficiencies, I would recommend to remove this part in the final version of the manuscript.

More minor concern

The authors refer to their approach as a non-genetic concentrating CO_2 approach which would be more or less opposed to a genetic approach. I find this somewhat misleading. Indeed, the effect of MOFs is only observed in the presence of a genetic CCM since no improvement is observed in high CO_2 grown algae. Also, the effect of MOFs is not to concentrate CO_2 , but rather to facilitate the functioning of the "genetic" CCM, likely by supplying more substrate to bicarbonate transporters. I would suggest the authors to define their system with a positive rather than negative term, using for instance "chemical approach" rather than as a "non-genetic approach".

Point-by-point response to the reviewers' comments

Reviewer #1 (Remarks to the Author):

In this new version the authors have addressed my concerns by performing additional experiments to better characterize the photosynthetic activity of algae associated or not with MOFs. The manuscript has been significantly improved, but there are still major improvements to be made.

Reply:

Thank you for your efforts in reviewing our manuscript and conferring on us professional comments to improve the manuscript. We have taken your comments seriously to provide more evidences for supporting the proposed scenario. All changes have been highlighted with yellow color in the revised version of the manuscript.

Comment 1

It is shown (Fig. 4C) that the apparent affinity of photosynthesis for C_i is somehow increased by the presence of MOFs when algae are grown in air but not when they are grown in high CO_2 . It is however difficult to evaluate the effect on the figure since the scale of the X-axis of the graph of Fig. S21 does not allow to distinguish lower CO_2 values (the scale should be extended between 0 and 0.5 to better evaluate the effect visually). The way errors bars shown on Fig. 4C and Fig. R1e,f is not explained and in the absence of statistical analysis, it is difficult to be convinced that the difference

observed in $K_{1/2}$ is significant. Moreover, these affinity determinations were performed using a light intensity of $1200 \mu\text{mol photons m}^{-2} \text{ s}^{-1}$ which is quite far from the light intensity used for the algal cultivation ($50 \mu\text{mol photons m}^{-2} \text{ s}^{-1}$). Altogether this makes the evidence quite weak to explain the observed effect on biomass productivity in terms of affinity. I would suggest the authors to discuss these points both in the result and in the discussion sections.

Reply:

Thank you for your comments. According your suggestions, on the one hand, we do the serial experiments at a proper light intensity for the algal cultivation ($50 \mu\text{mol photons m}^{-2} \text{ s}^{-1}$) and a reasonable series of C_i concentrations (mainly between 0 and 0.5 mM). On the other hand, to observe the details of the effect, we make the figures distinguished at lower CO_2 values. The results of the kinetic plots at the light intensity of $50 \mu\text{mol photons m}^{-2} \text{ s}^{-1}$ are similar to those of previous experiments using a light intensity of $1200 \mu\text{mol photons m}^{-2} \text{ s}^{-1}$, which is considered as saturated light for O_2 evolution. As shown in figure R1 and R2, because the initial kinetic plots of O_2 evolution dependence on the flux of incident light, the maximum O_2 evolution rates and the affinities for CO_2 are slightly changed. At lower CO_2 values, we clearly observe the different effects of affinities for CO_2 under different conditions. Error bars (Figures R1e, f and R2d) represent the standard deviation of $K_{1/2}$ values which are fitted and calculated by three independent experimental results. The way of error bars is explained in relevant legends of the figures. In a word, the results of the affinities for CO_2 conducted at the same light intensity as the algal cultivation ($50 \mu\text{mol photons m}^{-2} \text{ s}^{-1}$) provides the evidences to

support the observed effect on biomass productivity in terms of affinity. We have updated the results of the Figure R1 and R2 to the Fig. 4C in the revised manuscript and the Fig. S22, S27 in the supplementary information.

Figure R1 (Revised version of Supplementary Fig. 27). (a-d) The plot and the Michaelis-Menten fit of the net O_2 evolution rate versus the concentration of (a, c) CO_2 and (b, d) HCO_3^- for *C. pyrenoidosa* (Algae) and MOF/*C. pyrenoidosa* (MOF/Algae) cells grown under (a, b) air (LC, 0.04%) and (c, d) 2% CO_2 (HC) at pH 7. (e, f) The

Michaelis constant $K_{1/2}$ values as calculated from the Michaelis-Menten fit of the plot of the net O_2 evolution rate versus the concentration of (e) CO_2 and (f) HCO_3^- for *C. pyrenoidosa* (Algae) and MOF/*C. pyrenoidosa* (MOF/Algae) cells grown at pH 7 under air (LC, 0.04%) or 2% CO_2 (HC). Cell density of 1.0×10^7 cell mL^{-1} and the light source with a cutoff wavelength filter (under $50 \mu E m^{-2} s^{-1}$ irradiation, $\lambda > 600$ nm). Error bars in (e, f) represent the standard deviation of $K_{1/2}$ values as fitted from three independent experimental results.

Figure R2 (Revised version of Supplementary Fig. 22). (a-c) The plot and the Michaelis-Menten fit of the net O_2 evolution rate versus the concentration of CO_2 for *C. pyrenoidosa* (Algae) and MOF/*C. pyrenoidosa* (MOF/Algae) cells grown at (a) pH 6, (b) pH 7 and (c) pH 8 under air (LC, 0.04%). (d) The Michaelis constant $K_{1/2}$ values as calculated from the Michaelis-Menten fit of the plot of the net O_2 evolution rate

versus the concentration of CO₂ for *C. pyrenoidosa* (Algae) and MOF/*C. pyrenoidosa* (MOF/Algae) cells grown at different pH under air (LC, 0.04%). Cell density of 1.0×10^7 cell mL⁻¹ and the light source with a cutoff wavelength filter (under 50 μE m⁻² s⁻¹ irradiation, $\lambda > 600$ nm). Error bars in **(d)** represent the standard deviation of K_{1/2} values as fitted from three independent experimental results.

Comment 2

The authors also observe an increase intracellular accumulation of Rubisco in MOFs-treated algae, but do not really explain what could be the origin of such an effect and what could be its consequence in term of biomass productivity. Is there any evidence in the literature that Rubisco can limit photosynthesis under low CO₂ conditions? How is this additional Rubisco organized within the cell, for instance does it contribute to form larger pyrenoids?

Reply:

Thank you for your comments. Indeed, about the observation that intracellular accumulation of Rubisco is increased in MOFs-treated algae, we just report the results without explain what causes the effect. It is a key point of CO₂ fixation. In our scenario, MOFs-treated *C. pyrenoidosa* is more accessible to inorganic carbon sources under ambient air.

In the literature, Akiho Yokota *et al* ^[1] reported that the Rubisco content in *C. pyrenoidosa* was affected by the CO₂ concentration during cultivation. *C. pyrenoidosa* grown on high CO₂ concentration (1% CO₂) had higher Rubisco expression level of 1.5

mg (mg Chl)⁻¹ than that of 0.8 mg (mg Chl)⁻¹ in air-grown algae. Meanwhile, by comparing the Rubisco content and the photosynthetic CO₂ fixation rate, they found that nearly full activity of Rubisco must be needed during photosynthesis in *C. pyrenoidosa* and other green algae, which is different from that in C₃ plants. It implies the strong dependence of biomass growth rate on the Rubisco content in *C. pyrenoidosa*. Borkhsenius *et al* ^[2] applied immunogold particles (with anti-Rubisco antibody) to localize the distribution of Rubisco in green alga *Chlamydomonas reinhardtii* cells at different CO₂ concentrations. When 5% CO₂-grown algae were adapted to air-grown condition, the proportion of Rubisco in pyrenoid increased from 50% to 90% within 5 hours (other Rubisco was dissolved in the chloroplast stroma), but the Rubisco content in pyrenoid remained unchanged. The additional Rubisco is possibly distributed in the chloroplast stroma rather than assembled in the pyrenoid.

It indicates that the CO₂ concentration mainly affects the stroma-localized Rubisco content as shown in our work. Meanwhile, the starch sheath got thicker with the CO₂ concentration decreased, which indicated the induction of CCM to form the diffusive barrier for preventing the leakage of CO₂ from the pyrenoid ^[3].

Comment 3

It is not clear how the kinetic properties of Rubisco could be modified. Please discuss this point in relation to the current knowledge of Rubisco activase for instance.

Reply:

Thank you for your comments. The kinetic properties of Rubisco vary among different

photosynthetic organisms, but remain steady on a certain species ^[4]. Rubisco can be inactivated by mistaken generation of tight-binding sugar phosphates on the active sites, while Rubisco activase (Rca) regenerates the Rubisco catalytic site by release of the inhibitors after structural remodeling ^[5]. Mirkko Flecken et al ^[6] reported that Rca is positioned over the Rubisco catalytic site under repair and pulls the N-terminal tail of the large Rubisco subunit (RbcL) into the hexamer pore. Simultaneous displacement of the C terminus of the adjacent RbcL opens the catalytic site for inhibitor release. For the close-packed Rubisco in the pyrenoid of green alga, it seems not to be accessible for Rca like stromal Rubisco. Freeman Rosenzweig et al ^[7] observed the liquid-like behavior of the pyrenoid matrix in *Chlamydomonas reinhardtii* cell, enables the facile transportation of relatively insufficient Rca to enough Rubisco for efficient carbon fixation in the pyrenoid of green algae.

Based on the literature, we can reasonably infer that the enhanced Rubisco activity of MOF-treated *C. pyrenoidosa* is due to the increase of Rubisco content, but not the increase of the kinetic properties of Rubisco. According your suggestion, we have discussed the point in the revised manuscript.

Comment 4

The first paragraph of the discussion has no link with the current work. In connection with my concerns above, I would suggest that authors replace this section with a critical discussion of their own data. How can be the increase in biomass explained? What part is due to an increased affinity for CO₂ and what part could be due to other effects line

an increase in the Rubisco content, or a change in kinetics properties of Rubisco? How this relates to limitations previously reported in the literature?

Reply:

Thank you for your precious suggestion and opinion. We have rewritten the first paragraph of the discussion part in the revised manuscript. We critically discussed our data and cited the opinions in several literatures to explain the proposed scenario.

“In this study, we find that the affinity for CO₂ of MOF/*C. pyrenoidosa* cell is stronger than that of bare *C. pyrenoidosa* cell (Supplementary Fig. 27). But the presence of MOF particles doesn't remarkably have influence on the affinity of cells grown under high CO₂ (Supplementary Fig. 27). Because *C. pyrenoidosa* cells grown in high CO₂ is not harboring the intrinsic CCM [8]. The complete CCM is vital to transfer CO₂ toward Rubisco enzyme for accelerating the rate of CO₂ fixation. Moreover, we find that the expression level of Rubisco protein is upregulated in MOF/*C. pyrenoidosa* cells (Fig. 4e). It was reported that the Rubisco content in *C. pyrenoidosa* was affected by the CO₂ concentration during cultivation and nearly full activity of Rubisco must be needed during photosynthesis in *C. pyrenoidosa* and other green algae [1]. It was observed that Rubisco protein is more likely to stay in the chloroplast stroma than to be close-packed in the pyrenoid when CO₂ concentration is elevated. And the additional Rubisco was distributed in the chloroplast stroma rather than to be organized in the pyrenoid [2]. The kinetic properties of Rubisco vary among different photosynthetic organisms, but remain steady on a certain species [4]. It implies the biomass growth rate strong dependence on the Rubisco content in *C. pyrenoidosa*. Thus, the synergetic effect of the functional MOF and the intrinsic CCM in *C. pyrenoidosa* cells enables a strong affinity for CO₂ and the Rubisco content in *C. pyrenoidosa* for accelerating CO₂ fixation.”

Comment 5

Finally, I am very skeptical about the photosynthetic efficiency calculation (reaching values up to 7.1%), which is made by the authors taking into account the intensity of the incident light intensity and not taking into account light reflection, diffusion... Such calculation would need a careful determination of the light actually absorbed by cells (for instance using an integrating sphere) and cannot be only based on the incident light since light reflection at the surface of the cylinder and the light diffusion have significant contributions. Also, the nature of the light source should be taken into consideration. What is the spectrum of the light source used and what fractions of photons are absorbed by chlorophylls. See for instance Wagner et al. 2016 *Biotechnol. J* (<https://doi.org/10.1002/biot.201500357>) for the determination of photosynthetic conversion efficiency. Unless the authors would perform a more careful determination of the photosynthetic conversion efficiencies, I would recommend to remove this part in the final version of the manuscript.

Reply:

Thank you for your kind advice. Our previous photosynthetic efficiency calculation is not accurate and strict. Chemical energy fixed in biomass should not be estimated by the value of glucose molecule, the assessment of incident light energy should be accurate according to the nature of the light source. We carefully learn the literature you recommend to us and revised our calculation as below:

Apparent photo conversion efficiency calculations. The apparent photo conversion efficiency (APCE) values in this work were calculated according to *Wagner et al*^[9]. We define it as the conversion efficiency of the actual incident light to biomass:

$$\text{APCE (\%)} = E_B / E_I$$

E_B – energy fixed in biomass (J)

E_I – energy in actual incident light (J)

For the calculation of E_B , the calorific value (H_B) of 23.4 KJ g⁻¹ for *C. pyrenoidosa* [10] was taken to calculate the chemical energy stored in biomass. During two days cultivation, E_B can be calculated as:

$$E_B = v_B \times H_B \times V \times t$$

v_B – biomass growth rate (g L⁻¹ day⁻¹)

H_B – calorific value of biomass (J g⁻¹), here assumed to be 23.4 KJ g⁻¹

V – working volume of photobioreactor, 20 mL

t – process time, 2 days

For the calculation of E_I , a 7IGF10 grating spectrograph (Saifan Optoelectronic Instrument Co., Ltd., China) was used to measure the spectrum (Figure R4 and Fig. S15) of the light source. Photosynthetically active radiation (PAR) range of 400–700 nm is considered for the calculation of the average quantum energy of photons (U):

$$U = \int_{400nm}^{700nm} \frac{\phi(\lambda)hc}{\lambda} d\lambda \div \int_{400nm}^{700nm} \phi(\lambda) d\lambda$$

U – average quantum energy of photons (J)

$\phi(\lambda)$ – photon flux at different wavelength (count per second, arbitrary unit)

h – Planck's constant (6.626×10^{-34} J s)

c – speed of light (2.998×10^8 m s⁻¹)

λ – wavelength (nm)

A QSL-2101 radiometers (Biospherical Instruments Inc., USA) with an integrating

sphere was used to measure the actual incident light intensity (reflected and diffused light by photobioreactor and medium was excluded) in a photobioreactor filled with fresh BG-11 medium. The total energy in actual incident light E_I is:

$$E_I = N_A \times U \times I \times A \times t$$

E_I – energy in actual incident light (J)

N_A – Avogadro's constant ($6.022 \times 10^{23} \text{ mol}^{-1}$)

U – average quantum energy of photons (Joule per photon = J)

I – actual incident light intensity, measured to be $48.8 \mu\text{mol m}^{-2} \text{ s}^{-1}$

A – cross section area of algal suspension, measured to be 12 cm^2

t – process time, 2 days

Values were averaged over three independent experiments.

As a result, the apparent photo conversion efficiency (APCE) values are 5.1 % for the control group (bare *C. pyrenoidosa*) and 9.8 % for MOF/Algae hybrids.

Figure R3 (Revised version of Supplementary Fig. 15). The light spectrum of the light source for the cultivation of *C. pyrenoidosa*.

More minor concern

The authors refer to their approach as a non-genetic concentrating CO₂ approach which would be more or less opposed to a genetic approach. I find this somewhat misleading. Indeed, the effect of MOFs is only observed in the presence of a genetic CCM since no improvement is observed in high CO₂ grown algae. Also, the effect of MOFs is not to concentrate CO₂, but rather to facilitate the functioning of the “genetic” CCM, likely by supplying more substrate to bicarbonate transporters. I would suggest the authors to define their system with a positive rather than negative term, using for instance “chemical approach” rather than as a “non-genetic approach”.

Reply:

Thank you for your kind advice. We agree with your opinion that the role of MOFs is to assist the intrinsic CCM. We have changed all there “non-genetic” words to “chemical approach” in the revised manuscript.

Reference

1. A. Yokota, D. T. Canvin. Ribulose biphosphate carboxylase/oxygenase content determined with [¹⁴C]carboxypentitol biphosphate in plants and algae. *Plant Physiol.* **77**, 735-739 (1985).
2. O. N. Borkhsenius, C. B. Mason, J. V. Moroney. The intracellular localization of Ribulose-1,5-biphosphate carboxylase/oxygenase in *Chlamydomonas reinhardtii*. *Plant Physiol.* **116**, 1585-1591 (1998).
3. Z. Ramazanov, M. Rawat, M.C. Henk. The induction of the CO₂-concentrating mechanism is correlated with the formation of the starch sheath around the pyrenoid of *Chlamydomonas reinhardtii*. *Planta* **195**, 210-216 (1994).

4. C. Iniguez, S. C. Bauca, U. Niinemets. Evolutionary trends in RuBisCO kinetics and their co-evolution with CO₂ concentrating mechanisms. *Plant J.* **101**, 897-918 (2020).
5. M. A. Parry, A. J. Keys, P. J. Madgwick. Rubisco regulation: a role for inhibitors. *J. Exp. Bot.* **59**, 1569-1580 (2008).
6. M. Flecken, H. Wang, L. Popilka. Dual functions of a Rubisco activase in metabolic repair and recruitment to carboxysomes. *Cell* **183**, 457-4730 (2020).
7. E. S. F. Rosenzweig, B. Xu, L. K. Cuellar. The eukaryotic CO₂-concentrating organelle is liquid-like and exhibits dynamic reorganization. *Cell* **171**, 148-162 (2017).
8. J. H. Hennacy, M. C. Jonikas. Prospects for engineering biophysical CO₂ concentrating mechanisms into land plants to enhance yields. *Annu. Rev. Plant Biol.* **71**, 461-485 (2020).
9. I. Wagner, C. Steinweg, C. Posten. Mono- and dichromatic LED illumination leads to enhanced growth and energy conversion for high-efficiency cultivation of microalgae for application in space. *Biotechnol. J.* **11**, 1060-1071 (2016).
10. K. Sukačová, D. Búzová, P. Trávníček. Optimization of microalgal growth and cultivation parameters for increasing bioenergy potential: Case study using the oleaginous microalga *Chlorella pyrenoidosa* Chick (IPPAS C2). *Algal Res.* **40**, 101519 (2019).

REVIEWERS' COMMENTS

Reviewer #1 (Remarks to the Author):

I appreciate that the authors have taken my concerns seriously. Prior to acceptance, I think that the interpretation of data related to the increase in Rubisco levels in MOF-treated algae should be more careful.

The conclusion of the authors (lines 206-207) stating that "the acceleration of biomass growth is mainly due to upregulation of the expression level of the Rubisco..." is quite confusing. I understand from their previous experiments that the increase productivity at limiting CO₂ would result from a higher affinity for C_i due to the effect of MOF. The observed increase in the Rubisco content/activity may be a consequence, but in no case do the authors establish a link between this increase and the increased biomass productivity. Both effects may contribute, but the fact that the V_{max} values measured in MOF-treated algae are not increased (see Figs S22 & S27) suggests that the increase in Rubisco content does not improve the maximal capacity of CO₂ fixation. Therefore, I would suggest the authors to moderate their conclusion by mentioning that V_{max} values are not modified in MOF-treated algae, and write something more neutral like "such an increase in Rubisco content may result from an increased internal C_i concentration in MOF-treated algae, and may contribute to reaching high CO₂ fixation rates and high biomass productivity. Note however, that no increase in the maximal CO₂ fixation rate was observed non-limiting CO₂ (Fig S22, S27), indicating that the contribution of increased Rubisco content is probably limited in MOF-treated algae".

Line 62: please write "apparent photo conversion efficiency"

Line 64: please replace "on" by "by" the excreted carbonic anhydrase...

Line 209: please replace "stronger" by "higher" and correct the English in the whole paragraph.

Point-by-point response to the reviewers' comments

Reviewer #1 (Remarks to the Author):

I appreciate that the authors have taken my concerns seriously. Prior to acceptance, I think that the interpretation of data related to the increase in Rubisco levels in MOF-treated algae should be more careful.

Reply:

Thank you for your efforts in reviewing our manuscript and conferring on us professional comments to improve the manuscript. We have taken your comments seriously to provide more evidences for supporting the proposed scenario. All changes have been highlighted with yellow color in the revised version of the manuscript.

Comment 1

The conclusion of the authors (lines 206-207) stating that “the acceleration of biomass growth is mainly due to upregulation of the expression level of the Rubisco...” is quite confusing. I understand from their previous experiments that the increase productivity at limiting CO₂ would result from a higher affinity for C_i due to the effect of MOF. The observed increase in the Rubisco content/activity may be a consequence, but in no case do the authors establish a link between this increase and the increased biomass productivity. Both effects may contribute, but the fact that the V_{max} values measured in MOF-treated algae are not increased (see Figs S22 & S27) suggests that the increase in Rubisco content does not improve the maximal capacity of CO₂ fixation. Therefore,

I would suggest the authors to moderate their conclusion by mentioning that V_{max} values are not modified in MOF-treated algae, and write something more neutral like “such an increase in Rubisco content may result from an increased internal C_i concentration in MOF-treated algae, and may contribute to reaching high CO_2 fixation rates and high biomass productivity. Note however, that no increase in the maximal CO_2 fixation rate was observed non-limiting CO_2 (Fig S22, S27), indicating that the contribution of increased Rubisco content is probably limited in MOF-treated algae”.

Reply:

Thank you for your comments. Since the increased Rubisco content in MOF-treated algae results from the increased internal C_i concentration, the latter one should be the basic reason for the increased biomass growth rate in MOF-treated algae. We have replaced the sentence “the acceleration of biomass growth is mainly due to upregulation of the expression level of the Rubisco...” by more neutral explanation like “such an increase in Rubisco content may result from an increased internal C_i concentration in MOF-treated algae, and may contribute to reaching high CO_2 fixation rates and high biomass productivity. However, no increase in the maximal O_2 evolution rate was observed under non-limiting CO_2 (Supplementary Fig. S22, S27), indicating that the increased Rubisco content in MOF-treated algae mainly contributes to dark reactions”.

Minor questions

Line 62: please write “apparent photo conversion efficiency”;

Line 64: please replace “on” by “by” the excreted carbonic anhydrase;

Line 209: please replace “stronger” by “higher” and correct the English in the whole

paragraph.

Reply: Thank you for your kind advice. We have corrected all the mistaken words, and examined the text carefully.